# Immune stress suppresses innate immune signaling in preleukemic precursor B-cells to provoke leukemia in predisposed mice

Marta Isidro-Hernández [1,2,13], Ana Casado-García[1,2,13], Ninad Oak [3,13], Silvia Alemán-Arteaga [1,2,13], Belén Ruiz-Corzo[1,2], Jorge Martínez-Cano [4], Andrea Mayado[2,5], Elena G. Sánchez[6], Oscar Blanco[2,7], Ma Luisa Gaspar [8], Alberto Orfao [2,5], Diego Alonso-López [9], Javier De Las Rivas [2,10], Susana Riesco [11], Pablo Prieto-Matos [2,11], África González-Murillo [6], Francisco Javier García Criado[2,12], María Begoña García Cenador[2,12], Manuel Ramírez-Orellana [6], Belén de Andrés [8], Carolina Vicente-Dueñas [2,11,13] ✉, César Cobaleda [4,13] ✉, Kim E. Nichols [3,13] ✉ & Isidro Sánchez-García [1,2,13] ✉

The initial steps of B-cell acute lymphoblastic leukemia (B-ALL) development usually pass unnoticed in children. Several preclinical studies have shown that exposure to immune stressors triggers the transformation of preleukemic B cells to full-blown B-ALL, but how this takes place is still a longstanding and unsolved challenge. Here we show that dysregulation of innate immunity plays a driving role in the clonal evolution of pre-malignant $Pax5^{+/-}$ B-cell precursors toward leukemia. Transcriptional profiling reveals that $Myd88$ is down-regulated in immune-stressed pre-malignant B-cell precursors and in leukemic cells. Genetic reduction of $Myd88$ expression leads to a significant increase in leukemia incidence in $Pax5^{+/-}$Myd88$^{+/-}$ mice through an inflammation-dependent mechanism. Early induction of Myd88-independent Toll-like receptor 3 signaling results in a significant delay of leukemia development in $Pax5^{+/-}$ mice. Altogether, these findings identify a role for innate immunity dysregulation in leukemia, with important implications for understanding and therapeutic targeting of the preleukemic state in children.

Childhood B-ALL is considered a clonal blood disorder whose earliest stages take place during embryonic life. Hematopoietic progenitor cells carrying germline or somatic alterations affecting transcription factor genes that regulate early hematopoietic or early B cell development (such as germline *ETV6*, *PAX5*, or *IKZF1* mutations, or the somatic *ETV6-RUNX1* fusion) represent what are commonly defined as preleukemic cells[1]. These preleukemic cells are compatible with normal hematopoietic development in a large majority of cases. However, occasionally they can transform over time through the acquisition of additional somatic genetic alterations that lead to the appearance of full-blown B-ALL[2–5].

The nature of the environmental or intrinsic signals that contribute to this "switch" from the preleukemic state to the leukemic state remains unclear. The majority of preleukemic genetic alterations on their own are also insufficient to cause B-ALL in mice, essentially recapitulating the scenario in humans[2]. Recent preclinical studies have shown that exposure to an immune stressor such as common pathogens is necessary to provide the selection advantage that allows pre-leukemic cells to give rise to leukemia in genetically susceptible strains of mice[3,5]. Epidemiologic data demonstrating that infection can act as a trigger for B-ALL development in humans further supports these

findings[6–11]. However, the nature and order of the events by which infection triggers leukemia is still a matter of debate. The "delayed infection" hypothesis is based on the possibility that an immune system unexposed to common infections early in life may function abnormally when exposed to such infections later in life. Specifically, the abnormal immune signaling triggered by these infections is proposed to lead to a series of genetic events that culminate in the development of leukemia. Alternatively, the "population mixing" hypothesis was postulated to explain the appearance of small clusters of patients with B-ALL in a localized place and time[12]. It posits that childhood B-ALL arises as an uncommon result of exposure to any otherwise common infection, but this would only become noticeable as clusters of B-ALL at times of population mixing, when a relatively large number of susceptible children naïve to this infection mix with a large number of infected individuals, leading to a (largely subclinical) widespread epidemics[12]. In either case, a transient deficiency of the immune priming could system act together with a gut microbiome dysbiosis to drive B-ALL. However, little is known about how exposure to infection promotes such transformation of preleukemic cells.

Germline mutations affecting the B cell master regulator *PAX5* or the presence of the somatic *ETV6-RUNX1* fusion are associated with B-ALL development in a proportion of human carriers, and this process can be faithfully recapitulated in *Pax5*[+/−] or *ETV6-RUNX1* transgenic mice[3,5]. Using these and other mouse models, recent studies have begun to address how preleukemic cells respond to infectious stimuli; in this way, it has been demonstrated that infection-driven B-ALL development in *Pax5*[+/−] mice is T-cell independent[13]. Further, the presence of the *Pax5*[+/−] or *ETV6-RUNX1* mutations uniquely shapes the gut microbiota when compared to the microbiota of wild-type littermates[13]. Curiously, the *Pax5*[+/−]-specific gut microbiota protects these predisposed mice from leukemia development, and its alteration by antibiotic treatment early in life is sufficient to induce leukemia, even in the absence of exposure to common pathogens[13]. In the present study, we sought to determine how *Pax5*[+/−] and *ETV6-RUNX1* preleukemic cells switch to a leukemic state using the *Pax5*[+/−] and *ETV6-RUNX1* mouse models, which are uniquely suited for the study of the early stages of malignant transformation[2]. Herein, we describe how the temporary malfunction of innate immune signaling plays a key role in this process in *Pax5*[+/−] mice, and we identify how this is mediated by a molecular mechanism based on the partial downregulation of *Myd88*, finally leading to B-ALL in *Pax5*[+/−] mice. These findings have important implications for the understanding and potential therapeutic targeting of preleukemic cells in susceptible children.

## Results

### Microbiome disruption promotes leukemia development in *Pax5*[+/−] mice but not other genetically prone strains

Our group recently found that both *Pax5*[+/−] and *Sca1-ETV6-RUNX1* transgenic mice harbor distinct genotype-specific gut microbiomes directly related to the genetic alteration they are carrying[13]. This work established an experimental paradigm to evaluate and identify the factors that modulate the selective advantage of preleukemic hematopoietic precursors. Thus, we first asked whether this relationship between leukemia-predisposition genetic alteration and gut microbiome is an inherent characteristic of genetic alterations associated with infection-driven leukemias or, rather, predisposing genetic alterations associated with infection-independent leukemias. To test this possibility, we took advantage of the *Sca1-Lmo2* mouse model, which develops infection-independent T-ALL[14], and the *Sca1-BCR-ABLp190* model, which develops infection-independent B-ALL[15]. We initially analyzed the impact of these genotypes on the microbiome using stool samples from *Sca1-Lmo2* and *Sca1-BCR-ABLp190* mice. Indeed, the effect of each genetic alteration shaped a characteristic gut microbiome that was distinct between these two mouse strains and differed from the microbiome in wild-type (WT) and *Pax5*[+/−] animals

(Fig. 1a, b, Supplementary Fig. S1 and Supplementary Data 1–3). Thus, predisposing genetic alteration shapes distinct gut microbiota regardless of whether the alteration is associated with infection-dependent or infection-independent ALL development.

We have previously shown that altering the gut microbiome with antibiotic treatment elicits leukemia in *Pax5*[+/−] mice, even in the absence of an infectious trigger (i.e., in specific pathogen-free conditions, SPF). Therefore, we next explored whether altering the gut microbiome under SPF conditions might lead to leukemia induction in *Sca1-ETV6-RUNX1*, *Sca1-Lmo2*, and *Sca1-BCR-ABLp190* mice. These mice were treated with an antibiotic cocktail (ampicillin, vancomycin, ciprofloxacin, imipenem, and metronidazole) in their drinking water for 8 weeks, which was started at around 8 weeks of age. In all three models, leukemia incidence, time to disease onset and the disease characteristics from both immunophenotypic and histological points of view were not altered by antibiotic treatment compared to non-antibiotic treated mice (Fig. 1c, Supplementary Fig. S2). Thus, modifying the gut microbiome through bacterial depletion impacts B-ALL development in *Pax5*[+/−] mice but not in these other leukemia-predisposed strains. Taken together, these results suggest that, although childhood ALL up until now have been speculated to have a common underlying association with infection, there is etiologic heterogeneity regarding the role of gut microbiome dysbiosis in the development of different genetic subtypes of childhood ALL.

### Immune training by early exposure to infection does not prevent leukemia development in *Pax5*[+/−] mice

In agreement with the delayed exposure hypothesis, epidemiologic studies suggest that exposures leading to immune stimulation (i.e., breastfeeding, daycare attendance, contact with pets, allergies) during early childhood are correlated with reduced development of the most common B-ALL subtypes (*ETV6-RUNX1*[+] and high-hyperdiploid B-ALL)[16–21]. To ascertain whether the timing of infection exposure is relevant for B-ALL development in *Pax5*[+/−] mice, heterozygous mutant and control littermate WT mice were exposed to the same infectious conditions since pregnancy and birth, and then during the rest of their lifespan (we named this the "early exposure group") (Fig. 2a). The microbiological status in the conventional facility was defined and controlled for 2 years (Supplementary Data 4). The early exposure to common pathogens induced a significant decrease in peripheral blood (PB) B-cells in both WT and Pax5[+/−] mice. However, this decrease does not seem to be related to leukemogenesis as WT mice never develop B-ALL. Under this scenario, specific B-ALL development was observed in 25% (7 out of 28) of *Pax5*[+/−] animals (but none of the WT mice) of the early exposure group (Fig. 2b), closely resembling the penetrance of B-ALL development in *Pax5*[+/−] mice with delayed exposure[3] and in humans harboring a heterozygous *PAX5* c.547G>A pathogenic variant[22–24]. The leukemias occurring in *Pax5*[+/−] early exposure mice developed between 9 and 17 months of age, similar to those developing in delayed-exposure mice[3], and they exhibited the same leukemic immunophenotype (Fig. 2c and Supplementary Fig. S3–S5). Therefore, the timing of infection does not appear to impact leukemia development in *Pax5*[+/−] mice with exposure to an immune stressor (i.e., housing in a conventional, non-SPF, facility) representing the triggering event.

Nevertheless, it remained to be determined whether the mechanism driving B-ALL susceptibility in *Pax5*[+/−] mice under early exposure to infections was similar to the one driving B-ALL onset in *Pax5*[+/−] mice with delayed exposure. Examination of B cells in preleukemic *Pax5*[+/−] early exposure littermates revealed significantly reduced proportions of total B-cells in the peripheral blood (PB) when compared to WT littermates housed under similar conditions (Supplementary Fig. S6). This decrease in PB B-cell proportions was similar to what we observed in the delayed-exposure *Pax5*[+/−] group (Supplementary Fig. S6). To identify the somatically acquired second hits

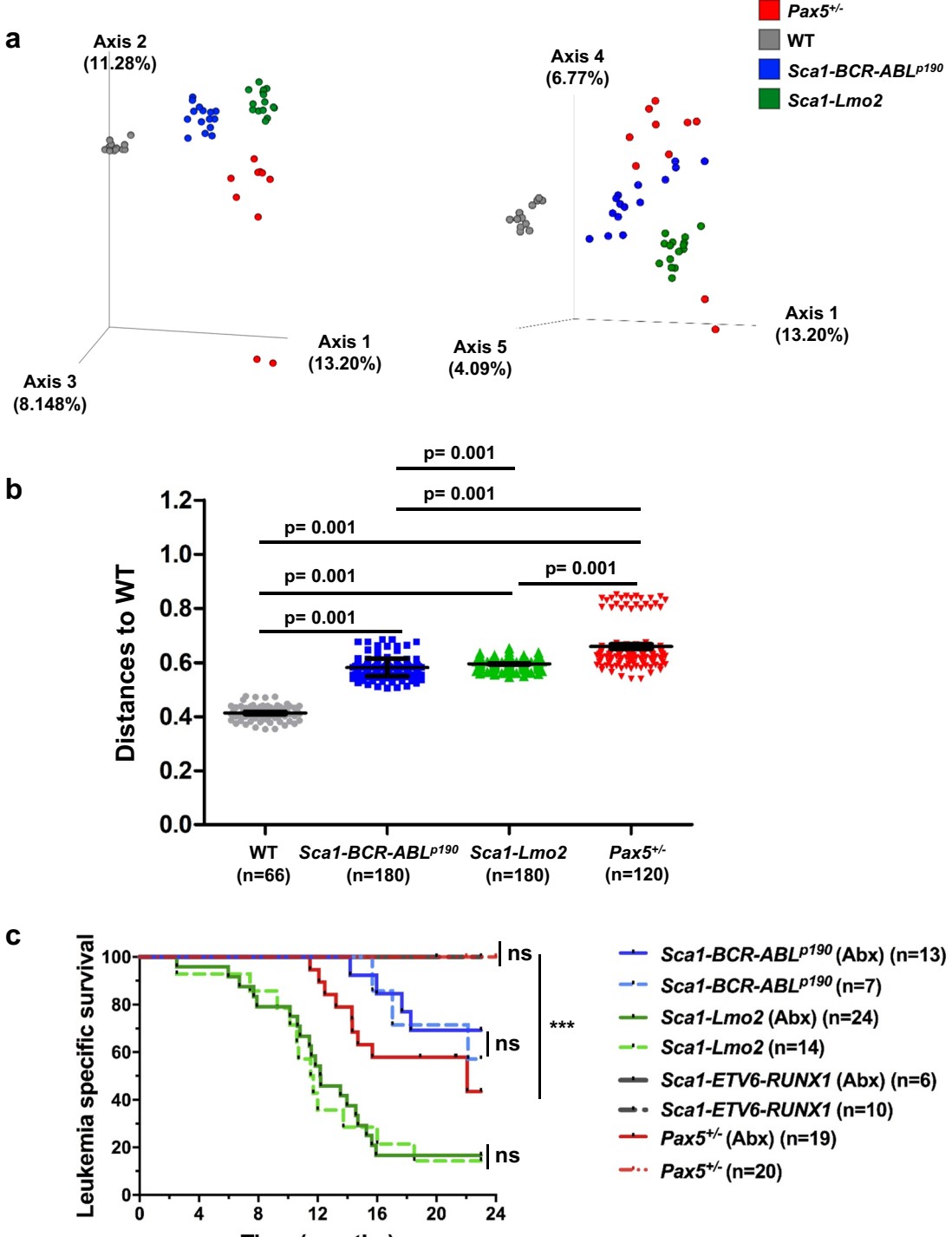

**Fig. 1 | Genetic susceptibility to ALL shapes a specific gut microbiota in pre-disposed mice. a** Pairwise Jaccard distances (beta diversity) were computed for all samples. Distance metric was ordinated into 3D via Principal Coordinates Analysis (PCoA) and visualized via Emperor. Axes indicate percent of explained variance. **b** Pairwise Permanova tests with 999 permutations were applied to test for differences in beta diversity grouped by mouse genotype (WT $n = 12$, *Sca1-BCR-ABLp190* $n = 15$, *Sca1-Lmo2* $n = 15$ and *Pax5+/−* $n = 10$). Scatter dot plots visualize Jaccard distances between samples of the same genotype. Data are shown as Mean ± SD, "$n =$" reports measured distances, "$p$" is the $p$-value of the Permanova test. **c** Leukemia specific survival curve of *Sca1-BCR-ABLp190* treated ($n = 13$) or untreated ($n = 7$) with antibiotic6s ($p$-value = 0.6514), *Sca1-Lmo2* treated ($n = 24$) or untreated ($n = 14$) with antibiotics ($p$-value = 0.7935), and *Sca1-ETV6-RUNX1* treated ($n = 6$) or untreated ($n = 10$) with antibiotics mice $p$-value > 0.9999), and *Pax5+/−* treated ($n = 19$) or untreated ($n = 20$) with antibiotics ($p$-value = 0.0002). None of them were exposed to common infections. *P*-values are from Log-rank (Mantel–Cox) test. Source data are provided as a Source Data file. Abx: treated with a cocktail of antibiotics for 8 weeks.

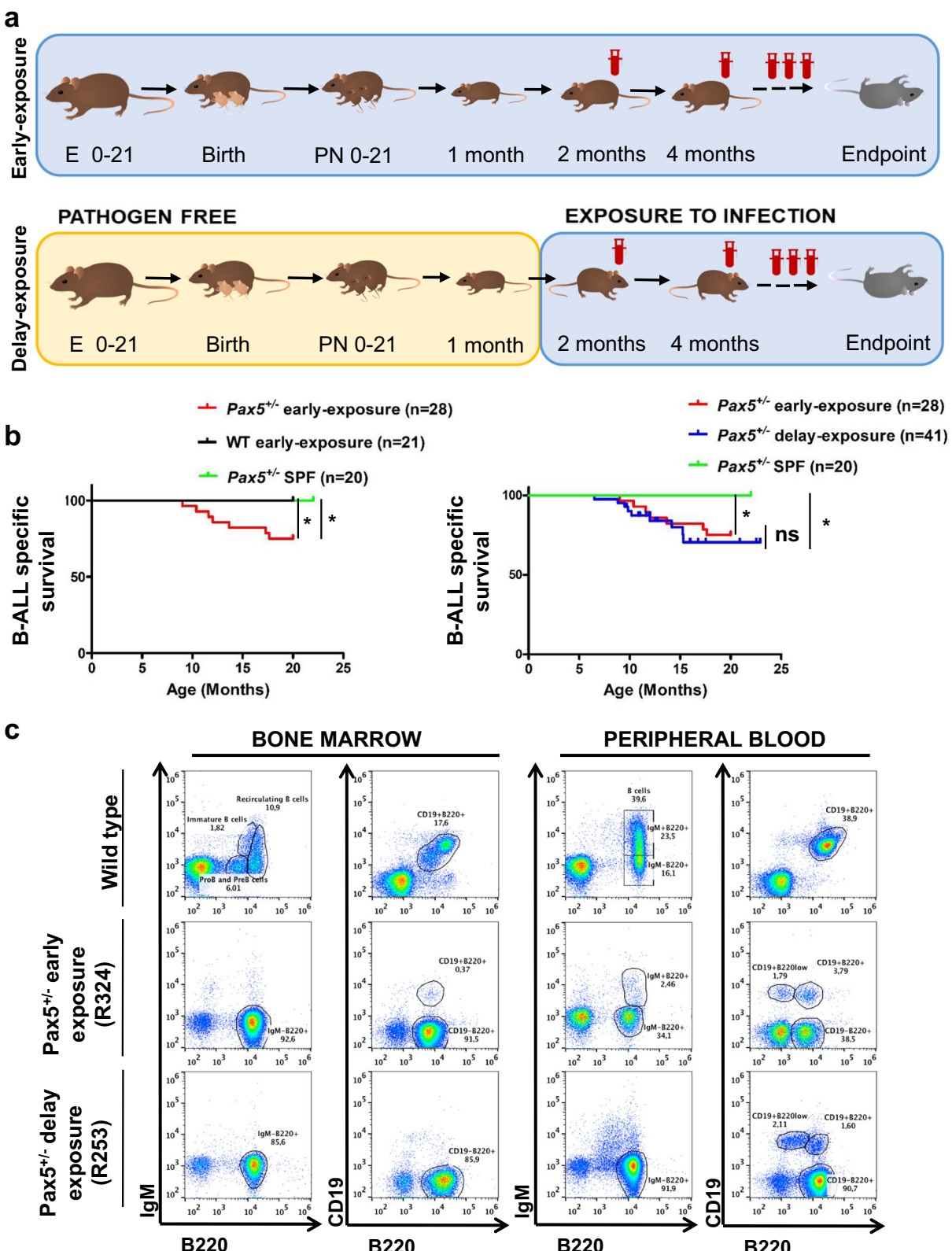

leading to leukemia development, we next performed whole exome sequencing of four *Pax5*+/− early exposure tumors and their corresponding germline. Similar to leukemias from the delayed *Pax5*+/− group, early exposure tumors showed recurrent mutations affecting the remaining WT *Pax5* allele, with Pax5 p.P80R occurring in 2 out of 4 analyzed leukemias (Supplementary Fig. S7). Moreover, an activating *Jak1* p.F837V variant was identified in one of the leukemic samples,

which again highlights the importance of the JAK/STAT-signaling pathway in *Pax5*+/− leukemias, as shown for *Jak1* and *Jak3* mutations in the *Pax5*+/− delayed-exposure B-ALL[3,25]. Interestingly, 2 out of 4 mice analyzed harbored somatic mutations affecting the RAS-RAF-MEK-ERK-pathway (Nras p.G13D and Kras p.Q61H) (Supplementary Fig. S7). Somatic mutations that deregulate this signaling pathway are among the most common genetic aberrations in childhood ALL[26]. In summary,

**Fig. 2 | Early exposure to infection promotes the development of B-cell precursor acute lymphoblastic leukemia in *Pax5*[+/−] mice. a** Experimental setup. Female mice of the "early exposure group" got pregnant already in a conventional facility (i.e., non-SPF conditions) where they are exposed to common infections during their lifespan, and the born pups were kept in these conditions for their lifetime. Female mice from the "delayed-exposure group" got pregnant under SPF conditions, and pups were kept in SPF until one month of age, when they were moved to the conventional facility. The mice were monitored periodically. The endpoint was established at 2 years of age (22–24 months) or when the mice showed signs of disease (clinical signs or presence of blasts in peripheral blood); the animals were sacrificed and the hematopoietic and non-hematopoietic tissues were analyzed. E: embryonic period; PN: postnatal period; SPF: specific pathogen free. **b** B-ALL specific survival curve of *Pax5*[+/−] animals born and kept into non-SPF conditions (red, *n* = 28) compared to WT control mice (black, *n* = 21) showing a significantly (log-rank *P* value 0.0151) shortened life span and compared to *Pax5*[+/−] animals always kept into SPF conditions (green, *n* = 20) showing a significantly (log-rank *P* value 0.0177) shortened life span (left panel), and B-ALL specific survival curve of *Pax5*[+/−] early exposure group (red, *n* = 28) compared to *Pax5*[+/−] delayed-exposure group (blue, *n* = 41) showing similar life span (log-rank *P* value 0.5491) and comparing *Pax5*[+/−] animals always kept into SPF conditions (green, *n* = 20) to *Pax5*[+/−] delayed-exposure group (blue, *n* = 41) showing a significantly (log-rank *P* value 0.0098) shortened life span (right panel). Source data are provided as a Source Data file. **c** Flow cytometric analysis of hematopoietic subsets in diseased *Pax5*[+/−] early exposure mice. Representative plots of cell subsets from the bone marrow and peripheral blood show accumulation of blast B cells in *Pax5*[+/−] early exposure mice (*n* = 7, age: 9–17 months) compared to control age-matched littermate WT mice (*n* = 4, age: 8–16 months). *Pax5*[+/−] early exposure mice show a similar immune-phenotype than *Pax5*[+/−] delayed-exposure mice (*n* = 9, age: 6–16 months).

we obtained in vivo evidence showing that *Pax5*[+/−] mice exposed to infections early in life develop leukemia with similar incidence, latency, and molecular features as *Pax5*[+/−] mice exposed later in life. These results show that immune training by early exposure to infection does not have a protective effect against the conversion of preleukemic cells to B-ALL in mice heterozygous for a germline *Pax5* mutation. These findings strongly point toward a cell-autonomous defect within *Pax5*[+/−] preleukemic cells and further reinforce the view of etiologic variation in leukemia development among different B-ALL molecular subtypes.

## Reduced expression of *Myd88* in immune-stressed preleukemic *Pax5*[+/−] precursor B-cells

Innate immunity plays a critical role in shaping the evolution of cancers with a demonstrated causal association with infection[27]. Indeed, it has been reported that, in infection-triggered cancers, Toll-like receptors (TLRs) play a central role in inhibiting as well as in promoting cancer formation, both at the level of immune cells within the tumor microenvironment and acting directly on the cancer cells themselves[27]. To elucidate whether TLR signaling is involved in the transition of pre-malignant B cells to B-ALL, we next performed a *Mouse Toll-Like Receptor Signaling Pathway* PCR Array using precursor B cells isolated from the bone marrow (BM) of *Pax5*[+/−] and WT mice kept either under SPF (Specific Pathogen Free) conditions or exposed to natural infection (i.e., housed in the conventional facility) both for 3 or 6 months. As we previously showed, *Pax5*[+/−] mice only developed B-ALL under natural infection exposure[3]. Analysis of the TLR signaling pathway in preleukemic precursor B cells of *Pax5*[+/−] mice exposed to infections revealed a decrease in the expression of the adaptors *Myd88* and *Tirap*, as well as the downstream transcription factor *Nfkb1a*, (Supplementary Fig. S8). Using real-time PCR quantification, we observed that low levels of *Myd88* could be detected in a small percentage of *Pax5*[+/−] preleukemic mice, and that this expression was within a similar range to that observed in WT precursor B-cells (Fig. 3a). However, *Myd88* expression was significantly decreased in *Pax5*[+/−] precursor B-cells from antibiotic-treated mice under SPF conditions and was equal to the level observed in *Pax5*[+/−] leukemic B-cells (Fig. 3a). The antibiotic treatment of ETV6-RUNX1 mice also induced a reduction in Myd88 expression in precursor B-cells (Fig. 3a). This finding suggests that the inability of Abx-treatment to drive leukemia progression in the *Sca1-ETV6-RUNX1* model therefore, does not appear to be due to a failure to down-regulate *Myd88* expression in precursor B-cells. While the *Sca1-ETV6-RUNX1* model might have a different reliance on Myd88 signaling or on the cell intrinsic developmental stage where molecular alterations should take place, these results highlight the need to identify downstream mechanisms to provide possible explanations for the inability of Abx-treatment to drive leukemia progression in the *Sca1-ETV6-RUNX1* model. Thus, induction of a leukemogenic state is associated with a decrease of *Myd88* expression in *Pax5*[+/−] B-ALL blasts.

## Downregulation of *Myd88* increases the penetrance of infection-driven B-ALL development in *Pax5*[+/−] mice

Given the downregulation of *Myd88* in preleukemic cells under antibiotic treatment as well as in *Pax5*[+/−] B-ALL blasts, we next directly tested the requirement for *Myd88* in B-ALL development. First, we decreased the expression of *Myd88* by generating *Pax5*[+/−];*Myd88*[+/−] mice by interbreeding, and examined whether *Pax5*[+/−];*Myd88*[+/−] animals were more prone to infection-induced B-ALL than *Pax5*[+/−] mice. *Pax5*[+/−];*Myd88*[+/−] mice were exposed to natural infections, and B-ALL development was monitored. We observed that B-ALL development was significantly increased in *Pax5*[+/−];*Myd88*[+/−] mice, where 64% (16 out of 25) developed B-ALL (Fig. 3b), closely resembling the penetrance of B-ALL development in *Pax5*[+/−] *animals* treated with antibiotics[13]. To extend these findings, SPF-housed, two months-old WT and *Pax5*[+/−] mice were intraperitoneally (i.p.) injected with PBS or 35 mg LPS (a Myd88-dependent TLR ligand) every other day for eight times over 2 weeks. This exposure of SPF-housed *Pax5*[+/−] mice to an Myd88-dependent TLR ligand generated B-ALL in 20% of the mice (Fig. 4). In contrast, the exposure of SPF-housed *Pax5*[+/−] mice to an Myd88-independent TLR ligand (mice treated with an intravenous injection of 200 μg of poly(I:C) for a total of 2 times, separated by four weeks between doses) did not generate B-ALL (Fig. 4), therefore supporting the direct participation of *Myd88* modulation in B-ALL development in this model. The advent of leukemia in *Pax5*[+/−];*Myd88*[+/−] mice under natural infection exposure occurred at between 7 and 20 months of age (Fig. 3b), similar to the *Pax5*[+/−] group. Thus, the latency in leukemogenesis was not changed. Furthermore, the critical role of *Myd88* downregulation for B-ALL development in *Pax5*[+/−] mice was further confirmed by both the lack of B-ALL development in *Pax5*[+/+];*Myd88*[+/−] mice under natural infection exposure (0 out of 26) (Fig. 3b) and the appearance of B-ALL in *Pax5*[+/−];*Myd88*[+/−] mice even in the absence of exposure to infections (5 out 8, 62.5%) (Supplementary Fig. S9), a condition where *Pax5*[+/−] mice never develop B-ALL[3]. In all these cases the characterization of *Pax5*[+/−];*Myd88*[+/−] B-ALLs showed that they are histologically and phenotypically similar to those appearing in the *Pax5*[+/−] group (Fig. 3c, d and Supplementary Figs. S10–S12).

To explore the relevance of our findings for human leukemia, we compared the role of *Myd88* activity on mRNA expression in from tumor cells from patients with B-ALL[28–31]. Using RNA micro-array gene expression analyses, we identified differentially expressed genes (*n* = 2463) in leukemic BM from *Pax5*[+/−];*Myd88*[+/−] mice compared with BM precursor B cells from WT mice (Supplementary Fig. S13 and Supplementary Data 5). When compared with defined signatures (genesets) previously identified as specifically up- or down-regulated in human B-ALLs[28,29,31], the transcriptional profiles of mouse B-ALLs behaved similar to the ones derived from human leukemias (Fig. 5a and Supplementary Fig. S13). Similarly, the genes differentially expressed between mouse B-ALLs and normal mouse B cell progenitors showed a parallelism with genesets related to malignant transformation (Supplementary Fig. S14). Therefore,

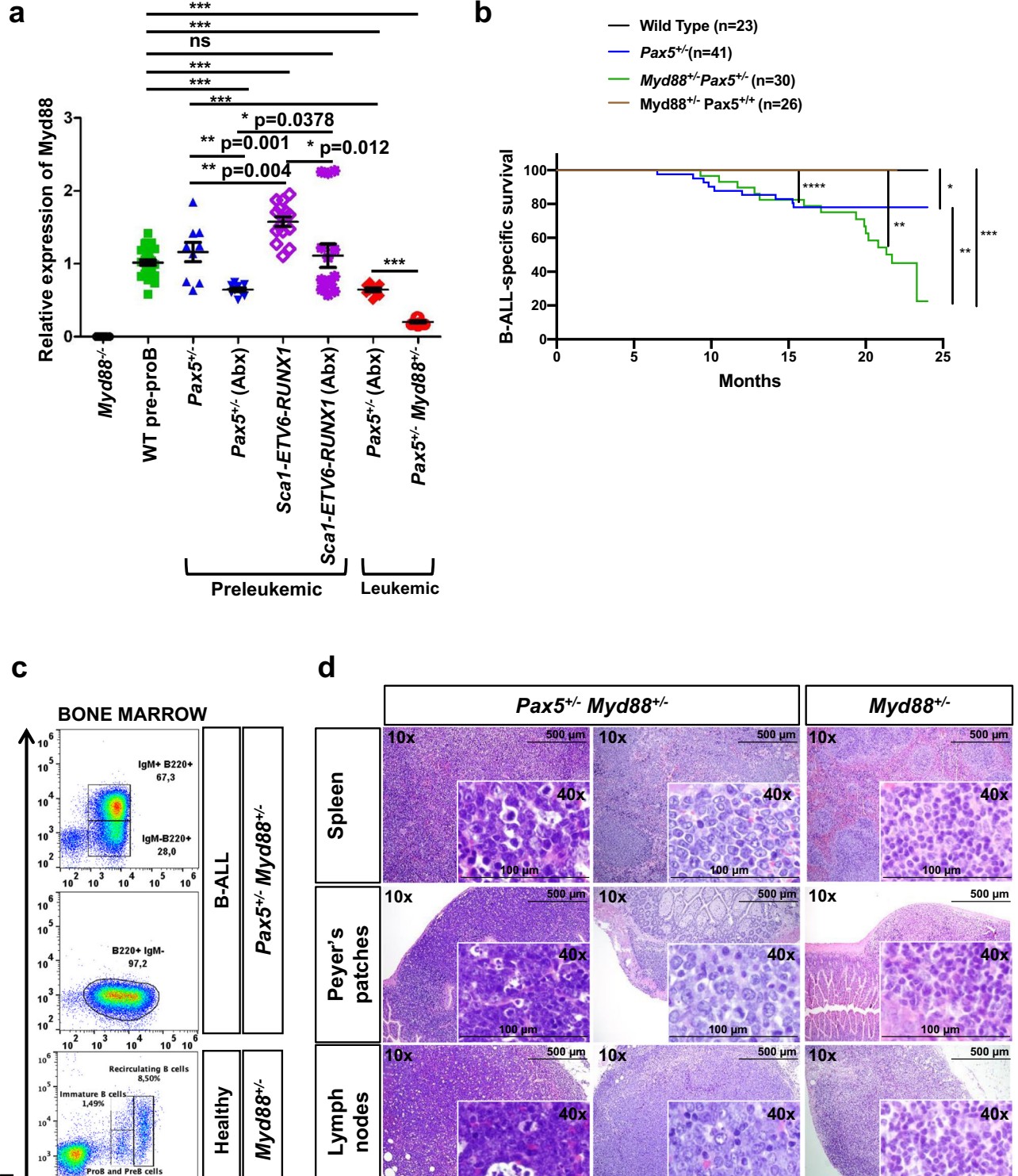

*Pax5*[+/−];*Myd88*[+/−] B-ALLs exhibited a gene expression pattern similar to human B-ALLs, and the gene expression profiles of *Pax5*[+/−];*Myd88*[+/−] and *Pax5*[+/−] B-ALLs did not show major differences between them or when compared with gene expression profiles derived from blast cells from patients with B-ALL (Supplementary Fig. S15 and Supplementary Data 6), indicating that *Myd88* is not significantly contributing to the leukemic gene expression profile of infection-driven B-ALL.

To identify somatically acquired second hits leading to B-ALL development in *Pax5*[+/−];*Myd88*[+/−] mice, we performed whole-genome sequencing of paired tumor and germline tail DNA samples from eight *Pax5*[+/−];*Myd88*[+/−] tumors. The percentage of leukemic cells in the sequenced samples ranged from 88.1% to 98.7%. We identified several somatically acquired recurrent mutations and copy-number variations involving B cell transcription factors in diseased *Pax5*[+/−];*Myd88*[+/−] mice (Fig. 5b, Supplementary Figs. S16, S17, Supplementary Data 7).

**Fig. 3 | Reduction in *Myd88* levels significantly increases the development of leukemias in *Pax5*[+/−] mice upon exposure to natural infections. a** Q-PCR of *Myd88* expression levels in preleukemic *Pax5*[+/−] and *Sca1-ETV6-RUNX1* and leukemic *Pax5*[+/−] proB cells. Preleukemic samples: sorted BM proB (B220[low] IgM[−]) cells from 3 months-old of wild-type (*n* = 5), *Pax5*[+/−] (*n* = 3), *Sca1-ETV6-RUNX1* (*n* = 5) and *Pax5*[+/−] (*n* = 3) and *Sca1-ETV6-RUNX1* (*n* = 5) mice treated with Abx for 8 weeks were analyzed by Q-PCR to quantified the expression of *Myd88*. Leukemic samples: BM leukemic cells from *Pax5*[+/−] mice treated with Abx for 8 weeks (*n* = 3) and *Myd88*[+/−]*;Pax5*[+/−] mice (*n* = 3) were analyzed by Q-PCR to quantify the expression of *Myd88*. All the mice analyzed were exposed to infections (in the conventional facility). Total BM from *Myd88*[−/−] mice was used as a negative control. Error bars represent the mean and SD. For the significant differences, unpaired *t* test *p*-values (two-tailed) are indicated in each case. Only relevant comparisons are shown. *p* = 0.0378 when comparing *Pax5*[+/−](Abx) vs *Sca1-ETV6-RUNX1*(Abx); *p* = 0.001 when comparing *Pax5*[+/−] vs *Pax5*[+/−](Abx); *p* = 0.012 when comparing *Sca1-ETV6-RUNX1* vs *Sca1-ETV6-RUNX1*(Abx); *p* = 0.004 when comparing *Pax5*[+/−] vs *Sca1-ETV6-RUNX1* and *** for *p*-value < 0.0001. **b** B-ALL-specific survival of *Pax5*[+/−] and *Myd88*[+/−]*; Pax5*[+/−] mice (*Pax5*[+/−], blue line, *n* = 41), (*Myd88*[+/−]*;Pax5*[+/−] dark green line, *n* = 30), (WT, black line, *n* = 23), and (*Myd88*[+/−]*;Pax5*[+/+] brown line, *n* = 26) all of them exposed to common infections. Log-rank (Mantel–Cox) test *p*-value > 0.9999 when comparing WT vs *Myd88*[+/−]*;Pax5*[+/+] mice, *p*-value = 0.0117 when comparing *Pax5*[+/−] vs *Myd88*[+/−]*;Pax5*[+/+] mice and *p*-value < 0.0001 when comparing *Myd88*[+/−]*;Pax5*[+/−] vs *Myd88*[+/−]*;Pax5*[+/+] mice. **c** Flow cytometric analysis of bone marrow showing the accumulation of blast B cells (B220[+] IgM[+/−]) in the leukemic *Myd88*[+/−]*;Pax5*[+/−] mice (X012 and W182) and compared with a healthy *Myd88*[+/−] mouse (X650), all of them exposed to common infections. **d** Haematoxylin and eosin staining of a tumor-bearing *Myd88*[+/−]*;Pax5*[+/−] mice showing infiltrating blast cells in the spleen, Peyer's patches, and lymph nodes, compared with an age-matched *Myd88*[+/−] mouse. Loss of normal architecture due to leukemic cell infiltration can be seen. A representative example of 16 diseased mice is shown in the figure. Magnification and the corresponding scale bar are indicated in each case. Source data are provided as a Source Data file.

Furthermore, recurrent mutations affecting the JAK/STAT and RAS signaling pathways were detected (Fig. 5b, Supplementary Fig. S16). The same genes are also mutated in *Pax5*[+/−] murine leukemias[3,13,25] and human B-ALL samples[26]. Thus, the drivers of infection-associated B-ALLs are similar in both *Pax5*[+/−]*;Myd88*[+/−] and *Pax5*[+/−] leukemic cells. Collectively, these observations revealed that downregulation of *Myd88* is associated with increased development of leukemia with a similar genetic profile as the leukemias observed in *Pax5*[+/−] mice and those from human patients.

## *Pax5*[+/−]*;Myd88*[+/−] precursor B cells exhibit an inflammatory transcriptional profile

We next studied the function of *Myd88* downregulation at the preleukemic stages by examining both the accumulation of myeloid cells and the transcriptional profiles of *Pax5*[+/−]*;Myd88*[+/−] preleukemic precursor B-cells in comparison with those from *Pax5*[+/−] preleukemic cells (Supplementary Figs. S18–21 and Supplementary Data 8–10). *Myd88* downregulation was not associated with an accumulation of myeloid cells in *Pax5*[+/−]*;Myd88*[+/−] mice when compared to *Pax5*[+/−]*;Myd88*[+/+] mice (Supplementary Fig. S18). However, *Pax5*[+/−]*;Myd88*[+/−] proB cells were characterized by an increased expression of genes involved in activation of signaling effectors (for example, Notch, apoptosis and IL2-STAT5 signaling) associated with cell proliferation and survival (Fig. 5c). *Pax5*[+/−]*;Myd88*[+/−] proB cells were also enriched in genes associated with pre-B and follicular murine B cell developmental stages (Supplementary Fig. S21A). Interestingly, the downregulation of *Myd88* in *Pax5*[+/−] proB cells exacerbated the number of genes differentially expressed that are regulated by Pax5, like *Cd19, Slamf6, Bst1, Foxo1*, or *Bcar3* (Supplementary Figure S21B). *Myd88* downregulation in *Pax5*[+/−] preleukemic precursor B-cells also augmented the expression of genes involved in the interferon-α response, TNFα signaling via NFκB and interferon-γ response (Fig. 5c), in agreement with the increase in inflammatory cytokines observed in peripheral blood in *Pax5*[+/−]*;Myd88*[+/−] mice (Supplementary Fig. S22). These findings suggest that Myd88-dependent signaling was hyperactive in *Pax5*[+/−] cells, and failed to upregulate feedback inhibitors of the Myd88–NFκB signaling pathway (Fig. 5c), in agreement with previous results showing that *Myd88* deficiency results in tissue-specific changes in cytokine induction and inflammation in infected mice[32]. These results are also in agreement with in vitro studies showing that pro-inflammatory cytokines like IL-6/IL-1β/TNFα[33] or TGF-β dependent signaling[34,35] promote malignant transformation of pre-leukemic B cells.

## Engagement of Myd88-independent TLR signaling delays natural infection-driven B-ALL development

We have shown that exposure to natural infections promotes full-blown B-ALL in *Pax5*[+/−] mice and that the malignant transformation of preleukemic precursor B-cells is associated with reduced levels of *Myd88* expression. Since Myd88 is a critical component in innate immunity induction following TLR ligation, we questioned whether Myd88-independent TLR ligation during the preleukemic phase might alter progression to B-ALL. Therefore, we have addressed this hypothesis by prior exposure of *Pax5*[+/−] mice to Myd88-independent TLR agonists, such as the synthetic double-stranded RNA analogue poly(I:C)[36]. Specifically, we established a mouse cohort consisting of 37 control littermate WT and 66 *Pax5*[+/−] mice. All mice were born in a SPF facility. Four weeks after birth, mice were transferred to a conventional facility, and 14 WT and 31 *Pax5*[+/−] mice were pre-treated with an intravenous injection of 200 μg of poly(I:C) for a total of 2 times, separated by four weeks between doses, and the remaining 23 WT and 35 *Pax5*[+/−] mice received vehicle (Fig. 6c). The microbiologic status of mice was monitored before and periodically throughout treatment, with no differences observed between treated and untreated animals (Supplementary Data 4). As a result of poly(I:C) treatment, we observed a specific increase in inflammatory cytokines and CD8[+] T cells in peripheral blood in both WT and *Pax5*[+/−] mice (Fig. 6b and Supplementary Figs. S23–S25), showing that poly(I:C) administration was having a similar immune-activating effect in WT and *Pax5*[+/−] mice. We next evaluated the immediate effects of Poly(I:C) treatment on the preleukemic cell population and myeloid cell compartments. Poly(I:C) treatment does not reduce the preleukemic compartment in the bone marrow of *Pax5*[+/−] mice (Supplementary Fig. S26) but it did induce an accumulation of myeloid cells in both WT and *Pax5*[+/−] mice (Supplementary Fig. S27). However, consistent with our previous findings, 23% (8 out of 35) of *Pax5*[+/−] mice receiving vehicle developed and succumbed to leukemia (Fig. 6c). The prevalence and the phenotypic and histological characteristics of B-ALL were similar in animals treated with poly(I:C) and untreated *Pax5*[+/−] mice, while 25% (8 out of 31) of treated *Pax5*[+/−] mice developed B-ALL and died from the disease (*p* = 0.7930) (Fig. 6c, and Supplementary Fig. S28, S29). In contrast, the latency to leukemia in *Pax5*[+/−] mice treated with poly(I:C) was significantly longer than in the *Pax5*[+/−] control group receiving vehicle, where it was delayed until 12–21 months of age (*p* = 0.0019) (Fig. 6e). Thus, early Myd88-independent TLR ligation during the preleukemic phase in *Pax5*[+/−] mice resulted in significant delay of B-ALL development under oncogenic environmental conditions, supporting the view that immune modulation during the preleukemic stage can significantly alter progression to B-ALL[2].

## Discussion

The association of B-ALL incidence with infectious stressors has been supported by epidemiological data for several decades now; however, until recently, there were no experimental models available capable of providing a suitable biological mechanistic explanation for these findings[3–5,13,25]. However, leukemogenesis in these animal models occurs relatively late related to its age peak in humans, and we do not

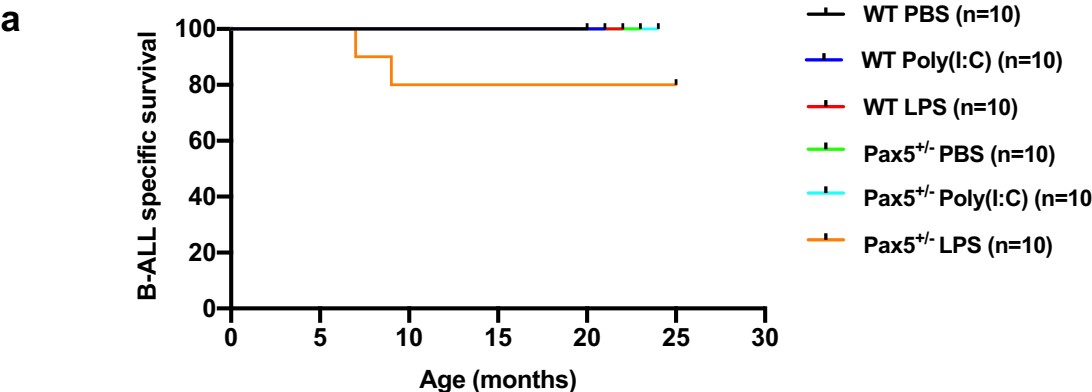

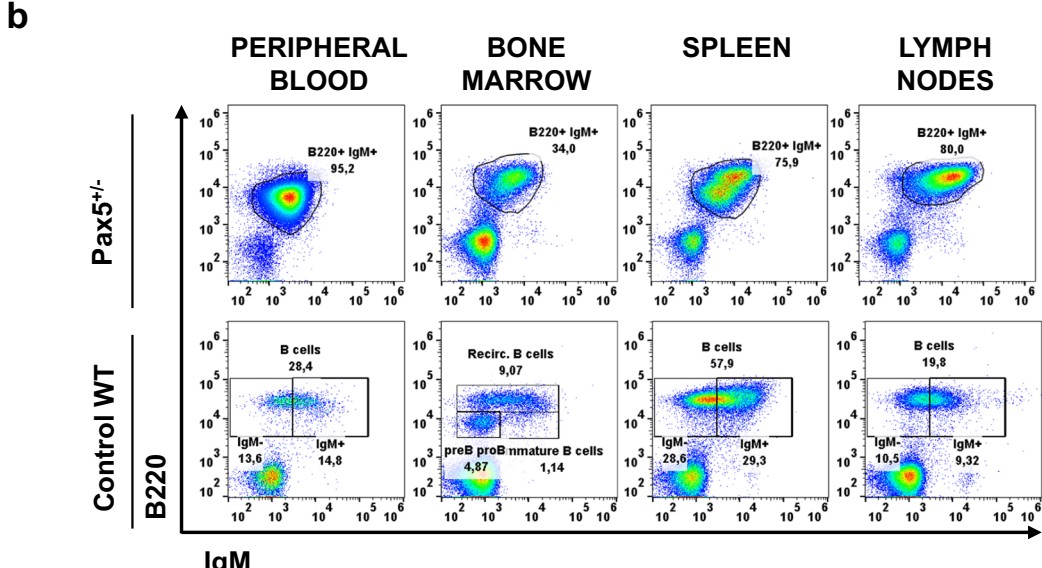

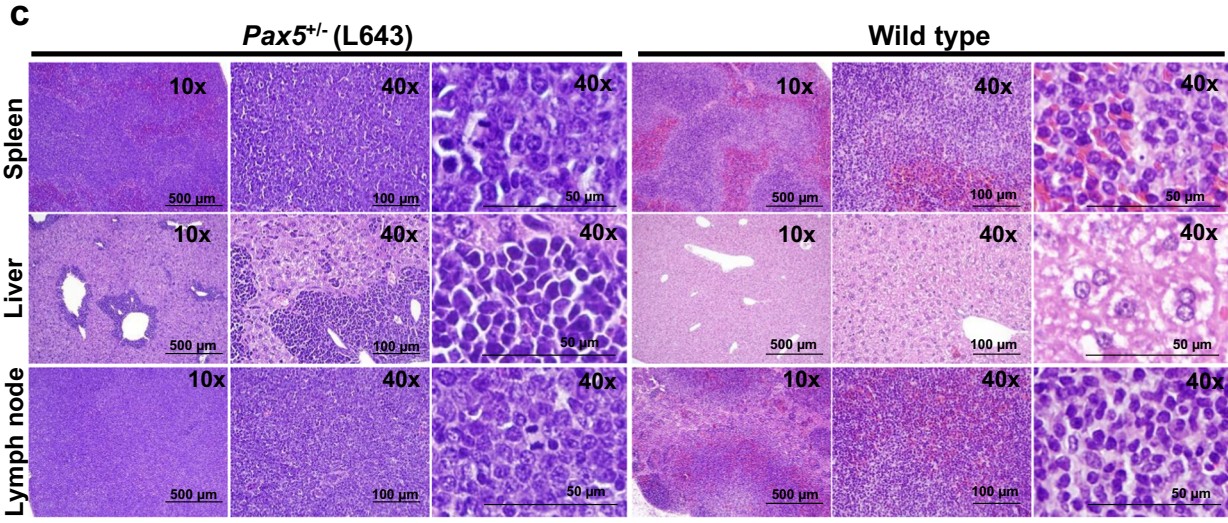

**Fig. 4 | *Pax5*⁺ᐟ⁻ mice injected with LPS develop B-ALL without infection exposure. a** B-ALL-specific survival of *Pax5*⁺ᐟ⁻ mice injected with LPS (a Myd88-depenedent TLR ligand) (orange line, *n* = 10), *Pax5*⁺ᐟ⁻ mice injected with poly(I:C) (a Myd88-indepenedent TLR ligand) (light blue line, n = 10), *Pax5*⁺ᐟ⁻ mice injected with PBS (as control) (green line, *n* = 10), control wild type (WT) mice injected with LPS (red line, *n* = 10), WT mice injected with poly(I:C) (dark blue line, *n* = 10), and WT mice injected with PBS (black line, *n* = 10), all of them housed in an SPF facility (without exposure to common infections). Log-rank (Mantel–Cox) test *p*-value = 0,1464 when comparing *Pax5*⁺ᐟ⁻ mice injected with LPS and *Pax5*⁺ᐟ⁻ mice injected with PBS. **b** Flow cytometry representative illustration of the percentage of leukemic B cells (B220⁺IgM⁺ subsets) in PB, BM, spleen and LN from a diseased *Pax5*⁺ᐟ⁻ mouse injected with LPS compared to an age-matched healthy WT mouse. **c** Haematoxylin and eosin staining of a tumor-bearing *Pax5*⁺ᐟ⁻ mouse injected with LPS unexposed to common infections showing infiltrating blast cells in the spleen, liver, and lymph nodes and compared with a healthy WT mouse. Loss of normal architecture can be seen due to the infiltrating cells morphologically resembling lymphoblast. Magnification and the corresponding scale bar are indicated in each case. Source data are provided as a Source Data file.

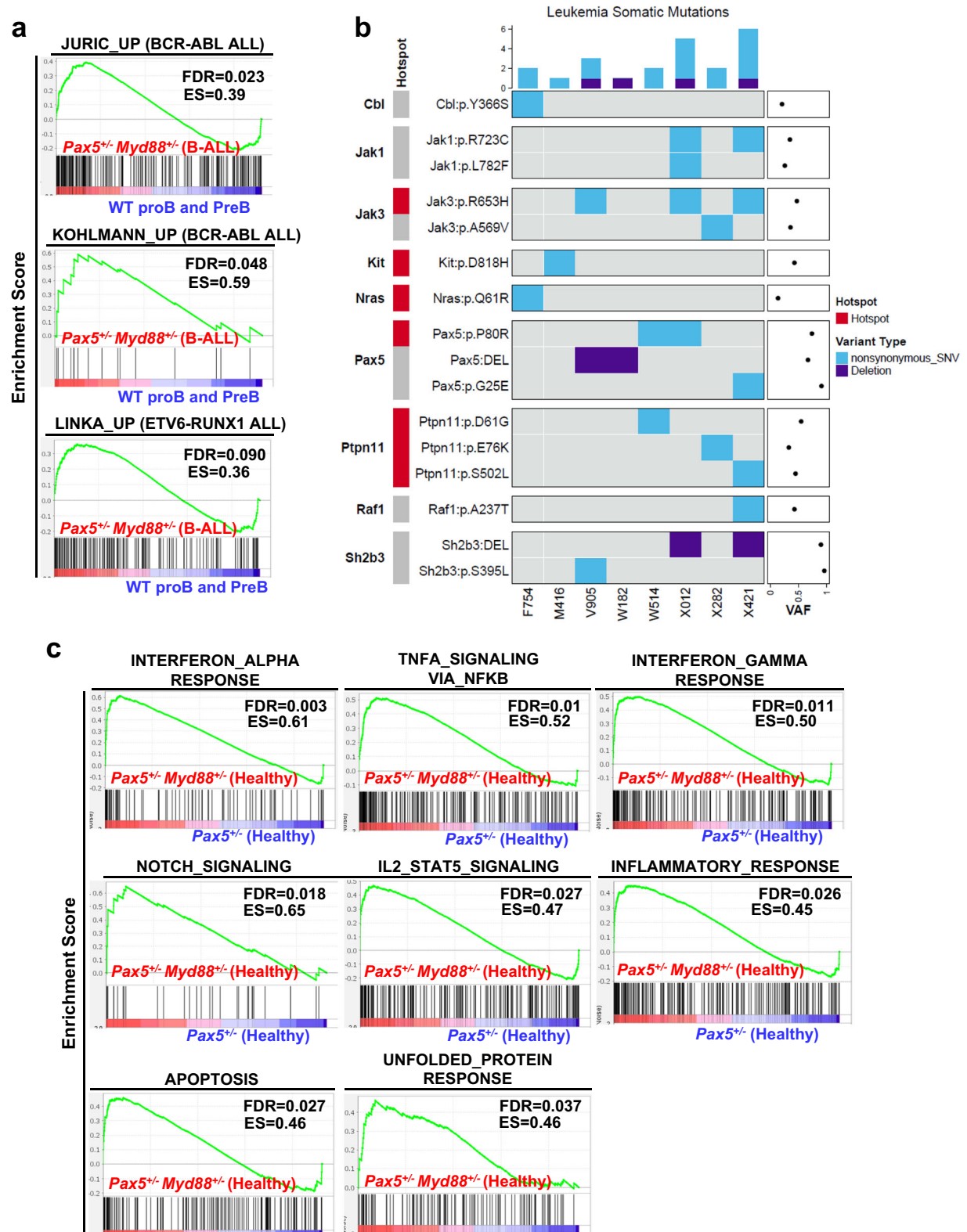

**Fig. 5 | Genetic impact of *Myd88*⁺/⁻ infection-driven leukemia. a** GSEA showing that leukemic *Pax5*⁺/⁻;Myd88⁺/⁻ cells from diseased mice present similar profiles for genesets previously identified in human B-ALL samples[28,29,31]. The genesets shown are significantly enriched with a nominal p-value < 1%. Statistical analysis was done using Gene set enrichment analysis (GSEA) algorithm. **b** Whole Genome Sequencing in *Pax5*⁺/⁻;Myd88⁺/⁻ B-ALLs. Oncoprint of somatic single nucleotide mutations and copy number alterations across 8 leukemia samples from *Pax5*⁺/⁻;Myd88⁺/⁻ mice. Somatic alterations are clustered by gene. Tumor DNA was derived from whole leukemic BM or LN, while tail DNA of the respective mouse was used as reference germline material. Previously reported known human or mouse leukemia hotspot mutations are highlighted (red). Mean tumor variant allele fraction (VAF) for each single nucleotide mutation is shown on the dotplot on the right. **c** GSEA plots showing the 8 gene sets that are significantly enriched at nominal p-value < 1% in *Pax5*⁺/⁻;Myd88⁺/⁻ proB cells from healthy mice. Enrichment analysis was done using the hallmarks collection from MSigDB[55]. Statistical analysis was done using Gene set enrichment analysis (GSEA) algorithm.

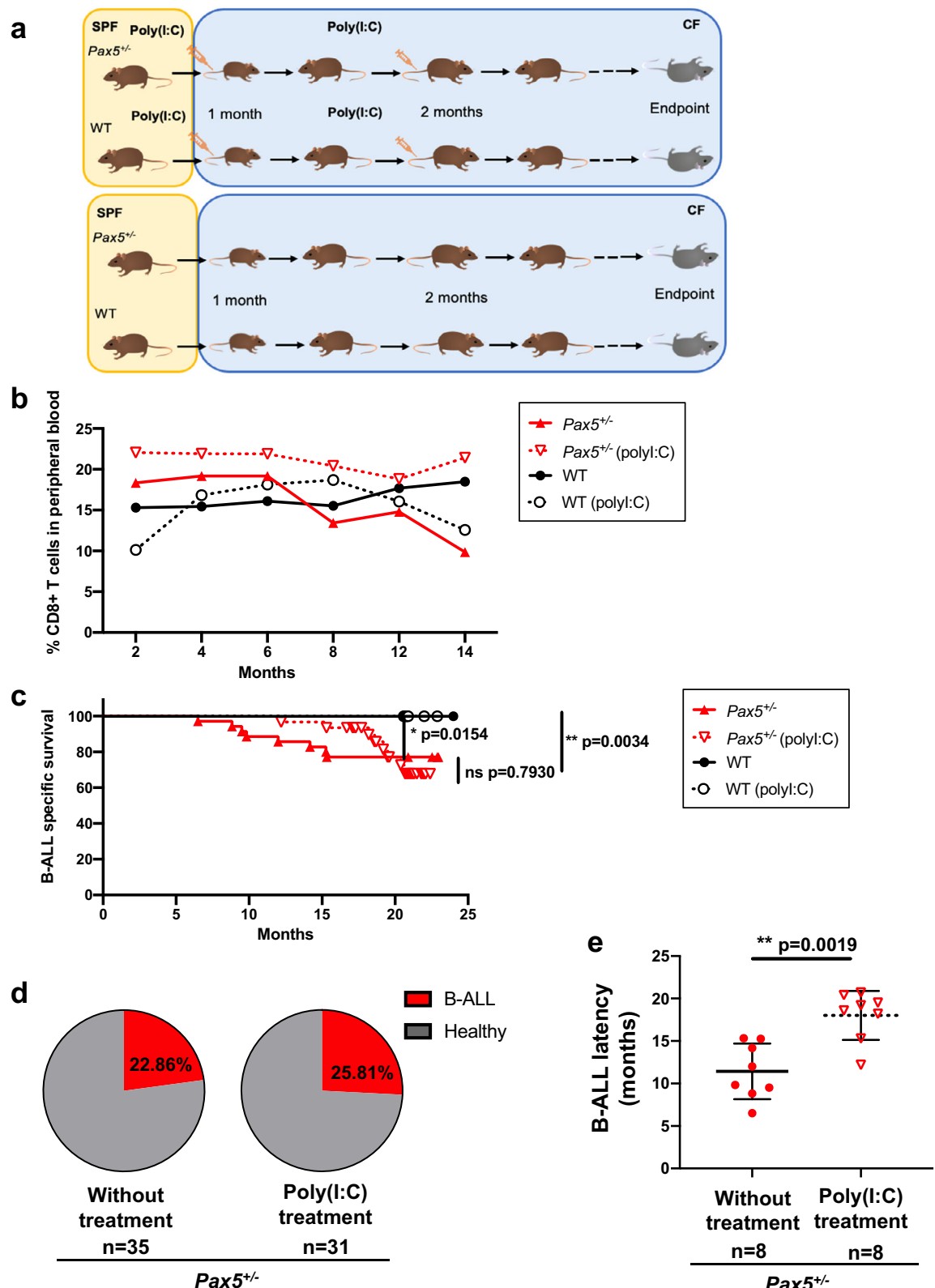

yet have an explanation for this finding[2]. We believe that the time to leukemogenesis is, to a large extent, related to the species-specific differences in the dependence of early B cell progenitors from IL7. Murine early B cell progenitors are strongly dependent on IL7 signaling, and when IL7 is removed from cell cultures, these cells die. IL7, when bound to its receptor, activates Jak3 and phosphorylates STAT5. We speculate that this may be the cause explaining why, under

the selection pressure of exposure to a common infectious environment, murine B cell progenitors need more time to acquire Jak3 mutations in comparison with their human counterparts, probably because human B cell progenitors are to a large extent independent of IL7[37].

Using these mouse models, here we have provided evidence showing that gut microbiome dysbiosis is a universal characteristic

**Fig. 6 | Innate stimulation by poly(I:C) treatment delays B-ALL development in Pax5[+/−] mice exposed to infections. a** Study design. poly(I:C) treatment fully described in methods section. SPF: specific pathogen free facility; CF: conventional facility, where the mice area exposed to common infections. **b** Percentages of CD8[+] T cells in peripheral blood of *Pax5[+/−]* and WT mice treated with poly(I:C) compared to untreated mice. An increase in CD8+ T cells can be observed in the blood of treated mice, only being statistically significant for the *Pax5[+/−]* mice (unpaired *t* test (two-tailed), *p*-value = 0.0087) (for WT mice the unpaired *t* test has a nonsignificant *p*-value = 0.5069). Each point represents the mean of the levels of the CD8+ T cells in all mice in each group for each of the different time points (*n* = 31 for *Pax5[+/−]* poly(I:C)-treated mice, *n* = 8 for *Pax5[+/−]* untreated mice, *n* = 14 for WT poly(I:C) treated mice and *n* = 23 for WT untreated mice). **c** B-ALL–specific survival of mice treated with poly(I:C) (*Pax5[+/−]*, pink dashed line, *n* = 31; WT, blue dashed line, *n* = 14) and nontreated mice (*Pax5[+/−]*, pink line, *n* = 35; WT, blue line, *n* = 23) following

exposure to common infections. Log-rank (Mantel−Cox) test *p* = 0.7930 when comparing *Pax5[+/−]* mice treated with poly(I:C) versus nontreated *Pax5[+/−]* mice, *p* = 0.0034 when comparing *Pax5[+/−]* versus WT poly(I:C)-treated mice and *p* = 0.0154 when comparing *Pax5[+/−]* versus WT mice without treatment. **d** Absence of changes in B-ALL incidence in *Pax5[+/−]* mice treated with poly(I:C). The treatment of *Pax5[+/−]* mice with poly(I:C) did not reduce nor increase the rate of B-ALL development. Fisher exact test (two-tailed), *p* = 0.9999. 8 out 35 mice in the non-treated group developed the disease (22.86%) and 8 out of 31 mice in the poly(I:C)-treated group developed B-ALL (25.81%). **e** poly(I:C) treatment increases the latency of B-ALL development in *Pax5[+/−]* mice. *Pax5[+/−]* untreated mice (*n* = 8) were diagnosed with B-ALL between 6 and 15 months of age and the *Pax5[+/−]* mice treated with poly(I:C) (*n* = 8) were diagnosed between 10.5 and 20.8 months of age. Error bars represent the mean and SD. Statistic test Mann−Withney (two-tailed), *p*-value = 0.0019. Source data are provided as a Source Data file.

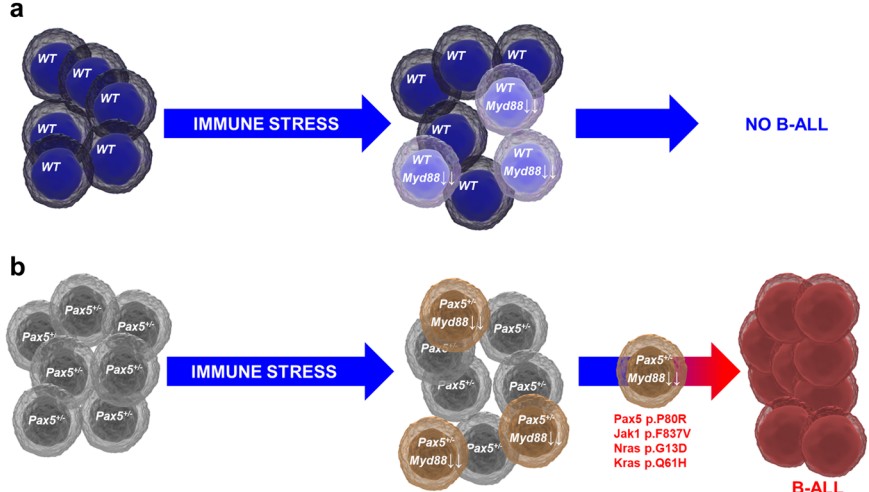

**Fig. 7 | Suppression of innate immune signaling by immune stress triggers leukemia development from *Pax5[+/−]* preleukemic precursors.** Immune stress (e.g. infection exposure, antibiotic treatment, etc.) can affect both healthy (**a**) and *Pax5[+/−]* preleukemic carriers (**b**) alike, and in both of them it can elicit a partial downregulation of *Myd88* expression in individual B cell progenitors. However,

only in *Pax5[+/−]* preleukemic progenitors will this downregulation interfere with innate immune signaling in a way that allows the appearance of secondary mutations leading to the development of a full-blown leukemia. The incidence of B-ALL is directly proportional to the percentage of *Pax5[+/−]* preleukemic cells having a downregulation of *Myd88* expression.

of all genetic preleukemic conditions tested. However, altering the gut microbiome through bacterial depletion exclusively has a specific impact on B-ALL development in genetically predisposed *Pax5[+/−]* mice, therefore highlighting the complex interaction between genetic subtypes and environmental factors that drive leukemogenesis in childhood B-ALL. Indeed, our study of variation in susceptibility to B-ALL by genetic subtypes has allowed us to identify for the first time an etiologic heterogeneity in the progression of the disease among subtypes of childhood B-ALL, and our results suggest that the use of gut microbiota as a modifiable therapeutic target to reduce or abolish the impact of immune stress in predisposed carriers may only apply to specific genetic subtypes of childhood B-ALL. In agreement with this finding, our studies show that, in *Pax5[+/−]* mice, immune training by early exposure to infection does not have a protective effect against the conversion of the preleukemic clone into B-ALL, therefore conflicting with the "delayed infection" hypothesis for this specific molecular subtype of B-ALL, and instead supporting Kinlen's "population mixing" explanation for the epidemiological findings. In this way, our results reinforce the notion of etiologic variation in childhood B-ALL development and the existence of a specific "gene/environment" interaction (i.e., infection) triggering B-ALL development.

Unexpectedly, our data also show that B-ALL development falls into the category of cancers associated with a dysregulation of the molecular mechanisms of innate immunity, which plays a central role

in the clonal evolution of pre-malignant B-cell precursors toward B-ALL as a result of immune stress. The molecular landscape characterization of innate immune regulators that promote the initiation of B-ALL has allowed us the identification of *Myd88* as one key player in this process. Indeed, a preleukemic B-cell-dependent role was evidenced by an increase in B-ALL incidence upon *Myd88* downregulation in the leukemia-prone *Pax5[+/−]* model (Fig. 7). The downregulation of Myd88 was shown to function in preleukemic B-cells by upregulating the expression of genes involved in interferon-α response, TNFα signaling via NFκB and interferon-γ response and activation of signaling effectors (for example, Notch, apoptosis and IL2-STAT5 signaling) associated with cell proliferation and survival. This inflammation-dependent role for *Myd88* in triggering B-ALL within preleukemic B-cells has also been shown for other TLRs in cancer[27]. The characterization of the innate immunity signaling within preleukemic precursor B cells showed that all *Pax5[+/−]* preleukemic cells are not identical (Fig. 6). As proposed, this variability in the expression of Myd88 under immune stress could provide a means for preleukemic cells to adapt their behavior to the environmental conditions and take advantage of potential fluctuations in the immune proficiency of the organism under stress conditions[2]. We speculate that such adaptation to new conditions might in turn lead to a rewiring of preleukemic cells themselves, which might affect immune molecules (such as TLRs and other immunomodulatory molecules), favouring the acquisition of second hits. Indeed, this

preleukemic heterogeneity could contribute to the variable disease risks across susceptible children.

The initial steps of B-ALL development in children pass usually undetected and, when the disease becomes evident at diagnosis, the delineation of the consecutive genetic events progressively leading to the leukemia is complicated by the accumulation of many oncogenic driver and passenger mutations. Therefore, key to the identification of the driver role of innate immunity dysregulation is the study of leukemia cells while they are appearing, in order to understand the sequential events initiating infection-driven B-ALL. This approach has allowed us to identify previously unknown molecular mechanisms taking place in the still unnoticed stages of human B-cell leukemogenesis, thus opening new opportunities for the development of effective strategies for the early detection and prevention of childhood B-ALL. Collectively, these data raise the possibility that innate immunity boosting might represent a viable strategy for risk reduction or prevention in childhood-related B-ALL; early innate immune response induction by Myd88-independent Toll-like receptor ligation during the preleukemic phase in *Pax5*+/- mice resulted in significant delay of B-ALL development under oncogenic environmental conditions, supporting the view that immune modulation during the preleukemic stage can significantly alter progression to B-ALL. TLR signaling induction produces disparate effects depending on the genetic subtype of childhood B-ALL and the experimental approach used. While early innate immune response induction by TLR ligation could reduce leukemia penetrance in the *Eµ-RET* and *E2A-PBX1* transgenic mouse models[38], ex vivo Toll-like receptor signaling induction by exposure to lipopolysaccharide triggers transformation of *ETV6-RUNX1*+ precursor B cells[39]. Our results also suggest that the elimination of these rewired preleukemic cells, or protection against this environmental adaption, might help curtail or prevent leukemic development in genetically predisposed carriers more effectively than strategies based on general immune training. Likewise, immune training by early exposure to infection does not prevent *Pax5*+/- preleukemic carriers against B-ALL. Overall, these findings have important implications for the understanding and potential therapeutic targeting of the preleukemic state in children.

## Methods

### Data reporting

Sample sizes were determined on the basis of the literature describing mouse modeling of natural infection-driven leukemia[3-5,13,25] and were justified by power calculations estimating 90% power to detect differential leukemia incidence. The experiments were not randomized and the investigators were not blinded to allocation during experiments and outcome assessment.

### Mouse model for natural immune stress-driven leukemia

*Pax5*+/-, *Sca1-ETV6-RUNX1*, *Sca1-Lmo2* and *Sca1-BCR-ABLp190* mice have been previously described[4,14,15,40]. The *Sca1-ETV6-RUNX1* construct was generated by placing the human *ETV6-RUNX1* cDNA under the control of the stem-cell-specific *Sca1* promoter, as has been previously described[4]. *Pax5*+/- mice were crossed with *Myd88*+/- mice (B6.129P2(SJL)-Myd88<tm1.1Defr>/J)[41] to generate *Pax5*+/-;*Myd88*+/- mice. These *Pax5*+/-;*Myd88*+/- mice were bred and maintained in the SPF area of the animal house until the moment when they were exposed to conventional pathogens present in non-SPF animal facilities, as previously described[3]. All mouse experiments were performed following the applicable Spanish and European legal regulations, and had been previously authorized by the pertinent institutional committees of both University of Salamanca and Spanish Research Council (CSIC). *Pax5*+/-(age of these mice range from 1 month-old until disease development), *Sca1-ETV6-RUNX1* (2 month-old mice were used in the study), *Sca1-Lmo2* (2 month-old mice were used in the study), *Sca1-BCR-ABLp190* (2 month-old mice were used in the study), and *Pax5*+/-;*Myd88*+/- mice (age of these mice range from 1 month-old until disease development) of a mixed C57BL/6×CBA background were used in this study. For the experiments, *Pax5*+/-, *Sca1-ETV6-RUNX1*, *Sca1-Lmo2* and *Sca1-BCR-ABLp190*, and *Pax5*+/-;*Myd88*+/- of the same litter were used. Both male and female mice were used in all the studies with the exception of the microbiome analysis in which only female mice were included in the study in order to avoid gender biases in microbiome results. Housing environmental conditions included a temperature of 21 °C ± 2°, humidity of 55% ±10%, and a 12 h : 12 h light: dark cycle. Mice had access to food and water *ad libitum*. During housing, animals were monitored daily for health status. When required, SPF-housed two months old WT or Pax5+/- mice were i.p. injected with PBS or 35 mg LPS corresponding to 35 EU (Escherichia coli 0111:B4, LPS-EB Ultrapure, Invivogen). When the animals showed evidence of illness, they were humanely killed, and the main organs were extracted by standard dissection. The maximal tumor size/burden approved by the institutional committees has not been exceeded in any mice included in the experiments. All major organs were macroscopically inspected under the stereo microscope, and then representative samples of tissue were cut from the freshly dissected organs, and were immediately fixed. Differences in Kaplan–Meier survival plots of transgenic and WT mice were analyzed using the log-rank (Mantel-Cox) test. Statistical analyses were performed by using GraphPad Prism v5.01 (GraphPad Software).

### Antibiotic treatment

Mice were treated with an antibiotic regimen as described[42]. Groups of mice were given a cocktail of antibiotics (ampicillin, 1 g/L, Ratiopharm; vancomycin, 500 mg/L, Cell Pharm; ciprofloxacin, 200 mg/L, Bayer Vital; imipenem, 250 mg/L, MSD; metronidazole, 1 g/L, Fresenius) added to their drinking water *ad libitum* for a period of eight weeks.

### poly(I:C) treatment

*Pax5*+/- and WT mice were born in SPF conditions and were maintained there until they reached 1 month of age. At that time, 31 *Pax5*+/- and 14 WT mice were transferred to a conventional facility (CF) and administered the first dose of poly(I:C) (intravenous injection of 200 µg/ mouse). As a control group, 35 *Pax5*+/- and 24 WT mice were transferred to CF and maintained under the same conditions without poly(I:C) administration. 4 weeks later, the second dose of poly(I:C) was administered to the previously treated animals. Mice were monitored periodically. The endpoint was established at 2 years of age (22–24 months) or when the mice showed signs of disease (clinical signs or presence of blasts in peripheral blood). SPF: specific pathogen-free facility; CF: conventional facility, where the mice area exposed to common infections.

### FACS analysis

Nucleated cells were obtained from total mouse bone marrow (flushing from the long bones), peripheral blood, thymus, or spleen. Contaminating red blood cells were lysed with red cell lysis buffer (RCLB) and the remaining cells were washed in PBS with 1% FCS. After staining, all cells were washed once in PBS and were resuspended in PBS with 1% FCS containing 10 µg/mL propidium iodide (PI) to excluded dead cells from both analyses and sorting procedures. The samples and the data were acquired in an AccuriC6 Flow Cytometer and analyzed using Flow Jo software. Specific fluorescence of FITC, PE, PI, and APC excited at 488 nm (0.4 W) and 633 nm (30 mW), respectively, as well as known forward and orthogonal light scattering properties of mouse cells were used to establish gates. Nonspecific antibody binding was suppressed by preincubation of cells with CD16/CD32 Fc-block solution (BD Biosciences). For each analysis, a total of at least 50,000 viable (PI-) cells were assessed. The gating strategy is exemplified in Supplementary Fig. S30.

The following antibodies were used for flow cytometry: anti-B220 (RA3-6B2, cat. 103212), CD4 (RM4-5, 1:250, cat. 100516), CD8a (53-6.7,

1:250, cat. 100708), CD11b/Mac1 (M1/70, 1:200, cat. 553310), CD19 (1D3, cat. 152404), CD117/c-Kit (2B8, 1:200, cat. 105807), Ly-6G/Gr1 (RB6-8C5; cat. 108412), IgM (R6-60.2, cat. 406509) and CD25 (PC61, cat. 553866) antibodies. All antibodies were purchased from BioLegend and used at a 1:100 dilutions unless otherwise indicated.

### Histology
Animals were humanely euthanized by cervical dislocation; tissue samples were formalin-fixed and embedded in paraffin. Pathology assessment was performed on hematoxylin-eosin stained sections under the supervision of Dr. Oscar Blanco, an expert pathologist at the Salamanca University Hospital.

### V(D)J recombination
V(D)J recombination analysis was carried out as previously described[43]. Immunoglobulin rearrangements were amplified by PCR using the primers below. Cycling conditions consisted of an initial heat-activation at 95 °C followed by 31–37 cycles of denaturation for 1 min at 95 °C, annealing for 1 min at 65 °C, and elongation for 1 min 45 s at 72 °C. This was followed by a final elongation for 10 min at 72 °C. To determine the DNA sequences of individual V(D)J rearrangements, the PCR fragments were isolated from the agarose gel and cloned into the pGEM-Teasy vector (Promega); the DNA inserts of at least ten clones corresponding to the same PCR fragment were then sequenced. The primer pairsused are listed in Table 1.

### 16S rRNA sequencing and analysis
**DNA extraction.** Fresh fecal samples were collected from each mouse (*WT* or *Pax5*+/− mice) at defined time points. Fecal DNA was isolated using QIAamp DNA Stool Mini Kit (Qiagen_Cat. No. 51604), following the stool pathogen detection protocol. DNA was extracted following the manufacturer's instructions with a modification: heating step in ASL buffer at 95 °C and eluting the DNA in 100 µl ATE buffer. Quantity of DNA was done by Picogreen (Invitrogen, cat.#P7589) method using Victor 3 fluorometry. After performing quality control, qualified samples proceed to library construction.

**Targeted 16S V4 region sequencing.** We targeted and amplified the V3-V4 region of the 16S rRNA gene by PCR using barcoded primers. Library construction was done by Macrogen (https://www.macrogen-europe.com/) using the Library Kit Herculase II Fusion DNA Polymerase Nextera XT Index Kit V2. To verify the size of PCR-enriched fragments, we checked the template size distribution by running on an

Agilent Technologies 2100 Bioanalyzer using a DNA 1000 chip. V3-V4 paired-end sequencing was performed using an Illumina MiSeq (La Jolla, CA) according to the manufacturer's protocols. The base calls (BCL) binary generated by the Illumina sequencer was converted into FASTQ files utilizing the Illumina package cl2fastq.

**Sequencing data processing and statistical analysis.** Microbiome bioinformatics were performed with QIIME 2 2019.10[44]. Raw sequence data were demultiplexed and quality filtered using the q2-demux plugin followed by denoising with DADA2[45] (via q2-dada2). All amplicon sequence variants (ASVs) were aligned with mafft[46] (via q2-alignment) and used to construct a phylogeny with fasttree2[47] (via q2-phylogeny). Alpha-diversity metrics (observed features and Faith's Phylogenetic Diversity[48]), beta diversity metrics (weighted UniFrac[49], unweighted UniFrac[50], Jaccard distance, and Bray-Curtis dissimilarity[51]), and Principle Coordinate Analysis (PCoA) were estimated using q2-diversity after samples were rarefied (subsampled without replacement) to 80992 sequences per sample. Taxonomy was assigned to ASVs using the q2-feature-classifier[52] classify-sklearn naïve Bayes taxonomy classifier against the SILVA 132 99% OTUs reference sequences[53]. For the comparative analysis of the taxonomy of the different groups, the feature table was filtered to keep features with a minimum frequency of 10 and present in at least 10 samples. The comparative analysis was determined using Linear Discriminant Analysis (LDA) Effect Size (LEfSe)[54].

### Mouse toll-like receptor signaling pathway PCR array
RNA from sorted BM precursor B (B220low IgM−) cells of *wild type*, and *Pax5*+/− was extracted with the TRIzol reagent (Invitrogen), and the cDNAs prepared with the RT2 First Strand kit (Qiagen) were pre-amplified using the RT² HT First Strand Kit cDNA cell cycle synthesis kit (Qiagen). The expression of TLR genes and their signaling pathways was analyzed in samples using the mouse RT² Profiler PCR Array (Qiagen). The PCR array DataanalysisV4web (Qiagen) source was used to calculate the threshold cycle (ΔCt) values and to obtain ΔCt values normalized to those of housekeeping genes. To determine the differences in relative expression between B cell populations, comparative analyses were performed ($2^{-\Delta\Delta Ct}$) using the PCR Array Data Analysis Software (Qiagen).

### Real-time PCR quantification (Q-PCR) of *Myd88*
Expression of *Myd88* was analyzed by Q-PCR in preleukemic samples (sorted BM precursor B (B220$^{low}$ IgM$^-$) cells of wild-type ($n = 3$), Pax5$^{+/-}$ ($n = 3$) and *Sca1-ETV6-RUNX1* ($n = 5$) and *Pax5*+/− ($n = 3$) and *Sca1-ETV6-RUNX1* ($n = 5$) mice treated with Abx for 8 weeks as well as in leukemic samples (leukemic cells from Pax5+/− mice treated with Abx for 8 weeks ($n = 3$) and leukemic cells from Myd88+/−;Pax5+/− mice ($n = 3$)). All of them exposed to infections (in mice housed in conventional facility). cDNA for use in quantitative PCR studies was synthesized using reverse transcriptase (SuperScript VILO™ cDNA Synthesis Kit; Invitrogen). Real-time PCR reactions were performed in an Eppendorf MasterCycler Realplex machine. Assays used for quantitative PCR are commercially available from IDT (Integrated DNA Technologies): Myd88 (Assay ID: Mm.PT.58.8716051.gs) and GAPDH (Assay ID: Mm.PT.39a.1). In addition, the probes were designed so that genomic DNA would not be detected during the PCR. Measurement of GAPDH gene product expression was used as an endogenous control. Total bone marrow from Myd88$^{-/-}$ mice were used as negative control. All samples were run in triplicate. The comparative CT Method (ΔΔCt) was used to calculate relative expression of the transcript of interest and a positive control. The change in threshold cycle (ΔCt) of each sample was calculated as the Ct value of the tested gene (target) minus the Ct value of GAPDH (endogenous control). The ΔΔCt of each sample was obtained by subtracting the ΔCt value of the reference from the ΔCt value of the sample. The ΔCt reference value used was the ΔCt

### Table 1 | Primers used for V(D)J recombination

| | | |
|---|---|---|
| V$_H$J558 | forward | CGAGCTCTCCARCACAGCCTWCATGCARCTCARC |
| | reverse | GTCTAGATTCTCACAAGAGTCCGATAGACCCTGG |
| V$_H$7183 | forward | CGGTACCAAGAASAMCCTGTWCCTGCAAATGASC |
| | reverse | GTCTAGATTCTCACAAGAGTCCGATAGACCCTGG |
| V$_H$Q52 | forward | CGGTACCAGACTGARCATCASCAAGGACAAYTCC |
| | reverse | GTCTAGATTCTCACAAGAGTCCGATAGACCCTGG |
| V$_H$Gam3.8 | forward | CAAGGGACGGTTTGCCTTCTCTTTGGAA |
| | reverse | GTCTAGATTCTCACAAGAGTCCGATAGACCCTGG |
| V$_H$3609 | forward | KCYYTGAAGAGCCRRCTCACAATCTCC |
| | reverse | GTCTAGATTCTCACAAGAGTCCGATAGACCCTGG |
| DH | forward | TTCAAAGCACAATGCCTGGCT |
| | reverse | GTCTAGATTCTCACAAGAGTCCGATAGACCCTGG |
| Cµ | forward | TGGCCATGGGCTGCCTAGCCCGGGACTT |
| | reverse | GCCTGACTGAGCTCACACAAGGAGGA |

M:A or C; S:C or G; R:A or G; W:A or T.

obtained from the total spleen of wild-type immunized mice. The fold change in each group, calculated as 2-ΔΔCt sample, was compared.

## Quantification of serum cytokine levels

Serum cytokine levels were analyzed using the Cytometric Bead Array (CBA) immunoassay systems (BD Biosciences), which assesses simultaneously IL6, IL10, MCP1, IFNg, TNFa e IL12p70 (CBA Inflammation Kit) and IL-2, IL-4, IL-6, IL-10, IL-17A, TNF alpha and IFN gamma in serum from the mice (Mouse Th1 Th2 Th17 Cytokine Kit #560,485; BDB). Briefly, 50 μL of the serum was incubated for 2 h at room temperature with 50 μL of anticytokine MAb-coated beads and 50 μL of the appropriate PE-conjugated anticytokine antibody detector. After this incubation period, samples were washed once (5 min at 200 g) to remove the excess of detector antibodies. Immediately afterward, data acquisition was performed on a FACSCanto II flow cytometer (BDB) using the FACSDiva software program (BDB). During acquisition, information was stored for 3,000 events corresponding to each bead population analyzed per sample (total number of beads >9000). For data analysis, FCAP Array Software v3.0 program (BDB) was used.

## Preleukemic pro-B cell culture

Pro-B cells were obtained from the BM of 2–3 weeks old WT, *Pax5*[+/-], and *Pax5*[+/-];*Myd88*[+/-] mice, respectively. B cells were isolated from the BM via positive MACS separation for B220. Cells were cultivated in special pro-B-cell medium (IMDM medium + L-glutamine + HEPES, 2% FCS, 1 mM penicillin–streptomycin, 0.03% Primaton, 50 μM beta-mercaptoethanol, 2 μg/ml cyproxin) supplemented with 5 ng/ml murine recombinant IL7 on ST2-feeder cells.

## Microarray data analysis

Total RNA was isolated in two steps using TRIzol (Life Technologies) followed by RNeasy Mini-Kit (Qiagen) purification, following the manufacturer's RNA Clean-up Protocol with the optional On-Column DNase treatment. The integrity and the quality of the RNA were verified by electrophoresis, and its concentration was measured. Samples were analyzed using Applied Biosystems Transcriptome Analysis Console (TAC) software v4.0.1 with the following parameters: (i) Analysis Version: version 1, (ii) Summarization Method: Gene Level−RMA, (iii) Genome Version: mm10 (Mus musculus) and (iv) Annotation: MoGene-1_0-st-v1.na36.mm10.transcript.csv. Differentially expressed genes were evaluated by specific statistical criteria ($p$-value < 0.05 and absolute fold change >2). Gene Set Enrichment Analyses were performed using GSEAv4.3.2[55–57], hallmark collection of gene sets[55], murine B cell developmental stages gene sets[58], and Pax5-regulated gene sets[59,60]. Gene expression signatures that are specifically upregulated or downregulated in human B-ALL[28–31] were also tested for enrichment within tumor specimens using GSEA, the list of gene sets is included in Supplementary Data 11.

## Whole-genome sequencing

Tumor DNA was derived from the BM where the percentage of blast cells was higher than 80% in each B-ALL and germline control DNA was obtained from the tail when the mice were 4-week old. Genomic DNA libraries were prepared from sheared DNA with the HyperPrep Library Preparation Kit (Roche PN 07962363001). Paired-end 150 cycle sequencing was performed on a NovaSeq 6000 (Illumina). Illumina paired-end reads were preprocessed and were mapped to the mouse reference genome (mm10) with BWA[61]. We used an ensemble approach to call somatic mutations (SNV/indels) with multiple published tools, including Mutect2[62], SomaticSniper[63], VarScan2[64], MuSE[65], and Strelka2[66]. The consensus calls by at least two callers were considered as confident mutations. The consensus call sets were further manually reviewed for the read depth, mapping quality, and strand bias to remove additional artifacts. Somatic copy-number alternations were determined by CNVkit[67]. For somatic structural variants (SV), four

SV callers were implemented in the workflow for SV calling, including Delly[68], Lumpy[69], Manta[70], and Gridss[71]. The SV calls passing the default quality filters of each caller were merged using SURVIVOR[72] and genotyped by SVtyper[73]. The intersected call sets were manually reviewed for the supporting soft-clipped and discordant read counts at both ends of a putative SV site using IGV.

## Sanger sequencing

Mutations were validated using Sanger sequencing on a 3130 Genetic Analyzer (Applied Biosystems). The following primer pairs were used: mJak3, *forward*: CGGGATGTGGGGCTTTAACT, *reverse:* GCAGACA CGGGGTATAGTGG; mJak1 *forward*: CCAGACAGCCAGGAGAACAG, *reverse:* CGTCTGCATAGTACCCACCC; mPax5 *forward:* CTCGTACATG CACGGAGACA, *reverse:* GGACCCTTCAGTACACCAGC.

## Statistical analysis

Statistical analyses were performed using GraphPad Prism v5.01 (GraphPad Software). Statistical significance was calculated by two-tailed unpaired $t$ test, paired $t$ test, Mann−Whitney test or Kruskal−Wallis test. Data normality was tested using the D'Agostino and Pearson test (alpha level = 0.05). Survival analyses were performed using the log-rank/Mantel−Cox test. Two-side Fisher exact test was used to compare leukemia incidence. $P$ values are depicted on the figures only for significant differences. According to power analysis calculations, assuming the development of leukemia as the experimental endpoint, and an expected proportion of 0.23 B-ALL-positive untreated animals at 20 months of age, a sample size of 34 in each group has a 73% power to detect a decrease in the incidence of the disease to a proportion of 0.03 B-ALL-positive mice in polyI:C-treated animals, with a significance level (alpha) of 0.05.

## Data availability

Authors can confirm that all relevant data are included in the paper and/or its supplementary information files. All data reported in this article are deposited in the NCBI's Gene Expression Omnibus (GEO)[74], available under the GEO Series accession number GSE221597 and to NCBI's Sequence Read Archive (SRA) under the BioProject accession number PRJNA912234. The remaining data are available within the Article, Supplementary Information or Source Data file. Source data are provided with this paper.

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

## Acknowledgements

The authors thank all the members of their laboratories for discussion. Research at C. Cobaleda's laboratory was partially supported by Ministerio de Ciencia e Innovación/AEI/FEDER (PID2021-122787OB-I00), and a Research Contract with the "Fundación Síndrome de Wolf-Hirschhorn o 4p-". Institutional grants from the "Fundación Ramón Areces" and "Banco de Santander" to the CBMSO are also acknowledged. Research in C. Vicente-Dueñas group has been funded by Instituto de Salud Carlos III through the project " PI22/00379 and by a "Miguel Servet Grant" [CPII19/00024-AES 2017-2020; co-funded by European Regional Development Fund (ERDF)/European Social Fund (ESF) "A way to make Europe"/"Investing in your future"]. Kim Nichols receives funding from the American Lebanese Syrian Associated Charities (ALSAC) and R01CA241452 from the National Cancer Institute Research in ISG group is partially supported by FEDER and by SAF2015-64420-R MINECO/FEDER, UE; by RTI2018-093314-B-I00 MCIU/AEI/FEDER, UE; by PID2021-122185OB-I00 MCIU/AEI/FEDER, UE, and by Junta de Castilla y León (UIC-017, CSI001U16, CSI234P18, and CSI144P20). Research in BdA group is partially supported by RTI2019-09114-B-I00 MCIU/AEI/FEDER, UE. M. Ramírez-Orellana and I. Sánchez-García have been supported by the Fundacion Unoentrecienmil (CUNINA project). C. Cobaleda, M. Ramírez-Orellana, and I. Sánchez-García have been supported by the Fundación Científica de la Asociación Española contra el Cáncer (PRYCO211305SANC). A. Casado-García (CSI067-18) and M. Isidro-Hernández (CSI021-19) are supported by FSE-Consejería de Educación de la Junta de Castilla y León 2019 and 2020 (ESF, European Social Fund) fellowship, respectively. S. Alemán-Arteaga is supported by an *Ayuda para Contratos predoctorales para la formación de doctores* (PRE2019-088887). J. Martínez-Cano is supported by a predoctoral fellowship FPI-UAM 2019. We also thank the Hartwell Center and Dr. Ti-Cheng Chang from the Center for Bioinformatics at SJCRH for their support of the whole genome sequencing and bioinformatic pipelines.

## Author contributions

Initial conception of the project was designed by M.R.-O., C.V.-D., C.C., K.E.N., and I.S.-G. Development of methodology was performed by M.I.-H., A.C.-G., N.O., S.A.-A., B.R.-C., J.M.-C., A.M., E.G.S., O.B., M.L.G., A.O., D.A.-L., J.D.L.R., S.R., P.P.-M., A.G.-M., F.J.G.C., M.B.G.C., M.R.-O., B.d.A., C.V.-D., C.C., K.E.N. and I.S.-G. O.B., M.B.G.C., F.J.G.C., and C.V.-D. performed pathology review. Management of patient information was performed by O.B., M.B.G.C., F.J.G.C., S.R., P.P.-M., M.R.-O., C.V.-D., K.E.N., and I.S.-G. M.I.-H., A.C.-G., N.O., S.A.-A., B.R.-C., J.M.-C., A.M., A.O., D.A.-L., J.D.L.R, S.R., P.P.-M., A.G.-M., F.J.G.C., M.B.G.C., M.R.-O., B.d.A., C.V.-D., C.C., K.E.N. and I.S.-G. were responsible for analysis and interpretation of the data (e.g., statistical analysis, biostatistics, computational analysis). Paper preparation was performed by M.I.-H., A.C.-G., N.O., S.A.-A., B.R.-C., J.M.-C., A.M., E.G.S., O.B., M.L.G., A.O., D.A.-L., J.D.L.R, S.R., P.P.-M., A.G.-M., F.J.G.C., M.B.G.C., M.R.-O., B.d.A., C.V.-D., C.C., K.E.N. and I.S.-G. Administrative, technical, or material support (i.e., reporting or organizing the data, constructing databases) was compiled by M.I.-H., A.C.-G., N.O., S.A.-A., B.R.-C., D.A.-L., C.V.-D., C.C., K.E.N., and I.S.-G. The study was supervised by M.R.-O., C.V.-D., C.C., K.E.N., and I.S.-G.

## Competing interests

O.B. reports personal fees from Takeda and Clinigen outside the submitted work. K.N. reports grants from Incyte outside the submitted work. No disclosures were reported by the other authors.

## Additional information

[1]Experimental Therapeutics and Translational Oncology Program, Instituto de Biología Molecular y Celular del Cáncer, CSIC-USAL, Campus M. de Unamuno s/n, Salamanca, Spain. [2]Institute of Biomedical Research of Salamanca (IBSAL), Salamanca, Spain. [3]Department of Oncology, St. Jude Children's Research Hospital, Memphis, TN, USA. [4]Immune system development and function Unit, Centro de Biología Molecular Severo Ochoa (Consejo Superior de Investigaciones Científicas -Universidad Autónoma de Madrid), Madrid, Spain. [5]Servicio de Citometría, Departamento de Medicina, Biomedical Research Networking Centre on Cancer CIBER-CIBERONC (CB16/12/00400), Institute of Health Carlos III, and Instituto de Biología Molecular y Celular del Cáncer, CSIC/Universidad de Salamanca, Salamanca, Spain. [6]Department of Pediatric Hematology and Oncology, Hospital Infantil Universitario Niño Jesús, Universidad Autónoma de Madrid, Madrid, Spain. [7]Departamento de Anatomía Patológica, Universidad de Salamanca, Salamanca, Spain. [8]Immunobiology Department, Carlos III Health Institute, 28220 Majadahonda (Madrid), Spain. [9]Bioinformatics Unit, Cancer Research Center (CSIC-USAL), Salamanca, Spain. [10]Bioinformatics and Functional Genomics Research Group, Cancer Research Center (CSIC-USAL), Salamanca, Spain. [11]Department of Pediatrics, Hospital Universitario de Salamanca, Paseo de San Vicente, 58-182, Salamanca 37007, Spain. [12]Departamento de Cirugía, , Universidad de Salamanca, Salamanca, Spain. [13]These authors contributed equally: Marta Isidro-Hernández, Ana Casado-García, Ninad Oak, Silvia Alemán-Arteaga, Carolina Vicente-Dueñas, César Cobaleda, Kim E. Nichols, Isidro Sánchez-García. ✉e-mail: cvd@usal.es; cesar.cobaleda@csic.es; Kim.Nichols@STJUDE.ORG; isg@usal.es

