## [Peer Review File · Nature Communications]

Immune stress suppresses innate immune signaling in preleukemic precursor B cells to provoke leukemia in predisposed miceREVIEWER COMMENTS

Reviewer #1 (Remarks to the Author):

In this manuscript, Isidro-Hernández and colleagues provide further description of the molecular changes associated with disease progression in their Pax5^{+/-} mouse model of acute lymphoblastic leukemia (ALL). Their previous studies with the Pax5 mouse have revealed intriguing associations between potential exposure to infection and/or loss of microbiota diversity with ALL progression. The resultant low penetrance disease resembles that of human ALL, suggesting it may be a relevant model for childhood B-ALL etiology.

Specifically, in this manuscript the authors show an association between Myd88 expression and leukemia progression, which correlates with an inflammatory transcriptional profile observed in B cell precursors from Pax5/Myd88 ^{+/-} mice. They further report the ability of non-Myd88 TLR signaling to delay leukemia onset in the Pax5 model. Notably, the data provided suggests that the influence of antibiotics on ALL incidence is limited to the Pax5^{+/-} model, an unexpected finding given the impact of leukemia-driving genomic aberrations on the microbiome complexity in other models.

Overall, this is an interesting study that provides a possible link between variables suspected to influence pediatric ALL onset. The indication that the drivers of leukemia progression may differ between ALL subtypes is relevant to the field, which has long speculated about a common underlying mechanistic association between infection and ALL. Despite the novel insights provided, however, the study has significant weaknesses in its current form, most notably the lack of appropriate control or comparator groups in the reported experiments.

Major concerns:

1. The effect of antibiotics in the additional models (Figure 1: BCR-ABL, Lmo2, and ETV6-RUNX1) is compared to that reported previously for Pax5^{+/-} mice; critically, no concurrently obtained results for the Pax5 model are presented. Given the significant impact reported for the poorly understood variables (i.e. SPF vs conventional housing) on disease progression, the potential that the conditions during the historical Pax5 study have changed cannot be dismissed. Without a demonstration that Abx-treated Pax5 mice developed leukemia in experiments performed in the same time frame as the additional models, the statement that it behaves differently is not sufficiently supported.
2. Similarly, in the early vs late exposure studies (Figure 2), survival of Pax5 mice maintained under SPF conditions over the same time period should be shown to confirm that the model retains the same need for infection and penetrance that was previously reported. These controls are essential while the precise nature of the environmental change driving leukemogenesis is unknown and thus cannot be shown to be consistent between experiments performed at different times.
3. While it is true that Myd88^{+/-} Pax5 mice show higher leukemia penetrance (Figure 3), it is an extremely late effect, only becoming apparent at a time when leukemia onset in Pax5^{+/-} mice appears to have ceased. Given this timing, it is essential to show survival curves for Myd88^{+/-} mice to show they do not develop leukemia after 20 months (which is not an unreasonable prediction given the inflammatory signature reported)? The need for this control is compounded by the claim (without supporting disease progression kinetics data) that leukemia occurs without “infection” in the double transgenics.

4. The reduction in Myd88 expression induced by antibiotic treatment of Pax5 mice is central to the conclusions of this study (Figure 3). It is surprising, therefore, that the same analysis was not performed in ETV6-RUNX1 mice, which also develops “infection-driven” ALL but does not show the same influence of antibiotic treatment. Such data would greatly inform the interpretation of the presented results and help to clarify the generality of such effects.

5. While the impact of Poly(I:C) on Pax5 leukemia development is interesting, the nature of the effect is not clear. Evaluation of acute effects of this treatment on the preleukemic cell population would be very informative. More importantly, the lack of any data on the effect of Myd88-mediated TLR signalling on Pax5[±] leukemia progression (in the presence and absence of infection exposure) is a significant omission, given the critical position in leukemogenesis proposed for this axis.

6. The study is presented in the context of the hypothesized causal role of infection on pediatric B-ALL, an etiology in which age of infection is thought to be important. However, leukemogenesis in the Pax5 model occurs relatively late (very late in the context of the higher penetrance in Pax5/Myd88 mice). This aspect of the model needs to be directly discussed.

Minor points:

“SPF (i.e., germ-free)” is not an accurate statement.

Supplementary Fig. S8B: The statement “... decrease in Myd88 expression was also observed in Sca1- ETV6-RUNX1 mice where the appearance of B-ALL is also triggered by infection exposure” does not appear to be supported by the data shown on the figure. Stats should be provided to show significance.

S14B: The description of GSEA results for Pax5[±], Pax5[±];Myd88[±], and human leukemia cells is either contradictory between the main text and the figure legend or needs to be rephrased for better clarity.

Reviewer #2 (Remarks to the Author):

The authors utilized the B-ALL development of genetically modified mouse models and examined the effect of infection on B-ALL development. They used several mouse models, such as Sca1-BCL-ABL, Sca1-Lmo2, and Sca1-ETV6-RUNX1. Although infection affects B-ALL development in Pax5[±] model, the infection did not affect B-ALL development in the three mice models used in this study. They tested early- and delay-exposure models and found no difference between the two models. Since MyD88 plays a key role in innate immunity, they measured MyD88 expression levels and examined the effect of MyD88 [±]-genetic background on B-ALL development. They also tested MyD88-independent stimulation using poly I:C. Unfortunately, this manuscript is not well organized, and several experiments are not performed appropriately. Thus, I am afraid that they failed to provide sufficient data to support their conclusions. Additionally, the underlying mechanism of the relationship between innate immunity and B-ALL development remains unclear. Considering

these weaknesses in this manuscript, This manuscript is too preliminary. Specific comments are described below.

Comment 1:

In Fig 1C, the authors showed that the Sca1-Lmo2 and Sca1-BCR-ABLp190 models developed B-ALL in a gut microbiota-independent manner, but it is unclear whether the Abx-treatment was sufficient. Pax5+/- should be included in Fig 1C experiment as a control. Alternatively, they should provide data showing that Abx-treatment was sufficient to exclude gut microbiota. In addition, statistical analysis should be performed to clarify whether the difference in leukemia-specific survival between Abx-treated and untreated Sca1-Lmo2 mice was significant or not.

Comment 2:

In Fig 2, the authors tested the early- and delayed-exposure groups. I am afraid that this experiment is a little confusing. There are many microorganisms even in SPF condition, and thus it is different from the pathogen-free condition. Moreover, Fig 2B lacks several important controls. In the left panel, Pax5+/- no exposure (pathogen-free condition) should be included as a control. In addition, the authors should investigate several mice and perform statistical analyses in Fig 2C.

Comment 3:

In Fig. 3, the authors investigate the expression of MyD88 and its relationship with B-ALL-specific survival. But the authors' explanation seems to be confusing. For instance, if the authors want to see the effect of MyD88, the experiments should be performed in MyD88 knockout background, not heterozygote.

Comment 4:

In Fig 5, the authors used poly I:C as a Myd88-independent ligand. It has been reported that poly I:C can be recognized by TLR3, RIG-I, and MDA5 in vivo. To test the effect, the authors should use TLR3, RIG-I, or MDA5 KO mice. Alternatively, MyD88-dependent ligands, such as LPS and peptidoglycan should be compared with poly I:C.

Comment 5:

Although they focused on innate immunity, the underlying mechanism observed in Figures 3 and 5 remains unclear. The authors should examine the innate immune responses, such as cytokine expression and accumulation of myeloid cells.

Minor comment 1:

The genetic constructs are unclear. For instance, the construct of Sca1-ETV6-RUNX1 should be explained sufficiently. In addition, the Sca1 construct is frequently used in this manuscript, but it is not so familiar.

Minor comment 2:

There is a little long introduction in the first paragraph of the result section. Most of those sentences should be included in the introduction.

Reviewer #3 (Remarks to the Author):

In this study, the authors Isidro-Hernández et al try to decipher the mechanistic role of Myd88 loss in promoting the onset of leukemia in Pax5+/- mice. This is an important model to study since the role of infection in the development of leukemia has been known for some time. But the exact pathogenesis is yet to be well laid out.

Overall, the paper is quite interesting and provides us with some novel insights about leukemogenesis in a Pax5+/- model. The immune upregulation after Myd88 knockdown appears quite significant and the poly I:C administration does slow down the onset of leukemia considerably. However, the paper does suffer from numerous concerns which are listed below.

Comments:

Figure 1A: The authors show that the microbiome of all the genotypes is different on the PCA plot. However, the explained variance is quite small in all the 3 axes. Can the authors also check for more principal components (PC4, PC5, etc.) to check whether they can explain the variance further?

Figure 1C: Can add the significance values between the pairs or NS to show that there is no difference between the antibiotic treated and non-treated groups in leukemia development. Even though there was no difference in survival, was there any difference in the histopathology of the tumors which developed? This may provide us with some clue about the differences in the microbiome contributing to phenotypic differences in the tumors.

Figure 1 demonstrates that disrupting the gut microbiome does not cause any change in leukemogenesis onset in different models. However, this is not investigated further. So, this figure is a little disjointed from the rest of the work.

Figure S1: The text is not legible.

Figure 2: The survival curves compare WT and Pax5+/- mice after early exposure to infections. Was Pax5+/- grown in SPF also used as a control? That would have been an ideal control to check for the phenotype which was intrinsic to the loss of one Pax5 allele with no infection exposure. Have the authors checked for the flowcytometry based immunophenotype of the hematopoietic tissues of Pax5+/- mice grow in SPF conditions?

The flowcytometry data shows an expansion of CD19-IgM-B220+ cells in Pax5+/- mice after infection exposure and supplementary figure S4 shows infiltrating blast cells in various organs. A higher magnification of the histopathology should be provided with the infiltrating malignant cells marked out. Did the authors find leukemic cells in the peripheral smear/bone smear examination? Those images can also be included.

Figures S4 and S6 would be served better by including a Pax5+/- control grown in SPF condition. There appears to be a difference between WT (early vs late exposure) and WT vs Pax5+/- for number of B220+ positive cells. Instead of ANOVA, the authors can use t-tests/Mann-Whitney to compare between two groups for significance.

Supplementary Figure S7: How was this experiment done? What was the source of the tumor tissue? Were cells sorted or purified in any way? What was the source of germline control? A detailed methodology would help in understanding the results better.

Supplementary Figures S8A-B: The error bars are quite high for most of the graphs and there does not seem to be any significant decrease in Myd88 expression in most conditions.

Can the authors clarify the exact significance levels and the reason for the variation?

Supplementary Figure S9: This is not a valid comparison since healthy marrow would have >70% of myeloid cells and their comparison with leukemic cells does not make much sense.

Figure 3: Were pre-pro B cells sorted from all samples? If so, what markers and methodology were used? The authors should clarify what they mean by 'pre-leukemic'? In fact, the text mentions precursor B-cells while the figure says pre-pro B cells. Myd88 is a critical component in innate immune signaling. Knockout of the same is associated with numerous abnormalities in the immune cell phenotyping. The authors should have used a Myd88^{+/-} Pax5 WT control which would have been ideal instead of a WT control for the flow cytometry and histopathology. The difference in survival between Pax5^{+/-} and the Pax5^{+/-}Myd88^{+/-} is not too different upto ~18 months and shows much after that. Am not sure how significant this would be for translational value.

The authors also mention that under SPF conditions, Pax5^{+/-} mice don't develop leukemias while the Pax5^{+/-} Myd88^{+/-} mice do. I could not find the data for the same (immunophenotyping, histopathology, etc.). Can the authors add the same since it is a significant result?

Figure 4: From the analysis in Figure 4A, were the authors able to identify which subtype of B-ALL did the tumors resemble transcriptomically? Some of the pathways identified in Supplementary Figure S13 (EMT, Myogenesis) may not be practically significant. The heatmap shows WT pre- and pro-B cells. What are these cells? The methodology is inadequate at many places and the terminology for the controls is used very interchangeably which makes it difficult to comprehend. A better experiment would be the comparison of the transcriptome of leukemic cells from Pax5^{+/-} Myd88 WT, Pax5 WT Myd88^{+/-} and Pax5^{+/-} Myd88^{+/-} mice.

In fact, the microarray and subsequent GSEA analysis can be more detailed (maybe in the supplementary data) mentioning the packages used to analyze, number of genes differentially expressed between all groups followed by the list of pathways significantly up/downregulated in the GSEA (using Hallmark/KEGG/GO pathways) with enrichment ratios. The present figures appear inadequate to explain the data from the experiment. Figure 4B: The authors mention that the mutations observed in Pax5^{+/-} tumors and Pax5^{+/-} Myd88^{+/-} tumors are not different. Did the authors perform any statistical analysis for the same?

Figure 4C and Supplementary Figure S19: The interpretations in the results section are too speculative. It is clear from Figure 4C that immune signaling pathways are upregulated after Myd88 knockdown in a Pax5^{+/-} background (Healthy Pax5^{+/-} Myd88^{+/-} vs. healthy Pax5^{+/-} Myd88 WT). However, the comparison in Figure S19 is not correct: WT vs. Pax5^{+/-} Myd88^{-/-}. There are too many variables in the KO and making conclusions that 'this inflammatory transcriptional profile was not present under complete loss of Myd88 in Pax5^{+/-} pro B cells, therefore being associated with intermediate, but not zero, levels of Myd88 expression' does not seem correct. The correct comparison would be Healthy Pax5^{+/-} Myd88 WT vs. healthy Pax5^{+/-} Myd88^{+/-} vs. healthy Pax5^{+/-} Myd88^{-/-} for meaningful conclusions.

Figure 5: Poly I:C treatment does not seem to cause much difference in CD8⁺ cell numbers as well as CD4⁺ and B-cells. In fact, the graphs intersect for the WT CD8⁺ cells. Can the

authors add statistics and strengthen their statement? The legend has too much methodology and results. Can shorten it considerably.

There is a clear difference in onset of B-ALL between the untreated and poly I:C treated groups. Was there any difference in the histopathology?

Supplementary Fig. S21: The figures are not legible. Paired t-tests would be appropriate in this case.

Supplementary Figs S22-23: Figures from Control Pax5^{+/-} mice can be added for comparison.

Minor comment: The manuscript would benefit from some spelling and grammar checks to fix multiple errors.

Point-by-point answer to the reviewer's comments:

Reviewer # 1:

In this manuscript, Isidro-Hernández and colleagues provide further description of the molecular changes associated with disease progression in their Pax5^{+/-} mouse model of acute lymphoblastic leukemia (ALL). Their previous studies with the Pax5 mouse have revealed intriguing associations between potential exposure to infection and/or loss of microbiota diversity with ALL progression. The resultant low penetrance disease resembles that of human ALL, suggesting it may be a relevant model for childhood B-ALL etiology.

Specifically, in this manuscript the authors show an association between Myd88 expression and leukemia progression, which correlates with an inflammatory transcriptional profile observed in B cell precursors from Pax5/Myd88 ^{+/-} mice. They further report the ability of non-Myd88 TLR signaling to delay leukemia onset in the Pax5 model. Notably, the data provided suggests that the influence of antibiotics on ALL incidence is limited to the Pax5^{+/-} model, an unexpected finding given the impact of leukemia-driving genomic aberrations on the microbiome complexity in other models.

Overall, this is an interesting study that provides a possible link between variables suspected to influence pediatric ALL onset. The indication that the drivers of leukemia progression may differ between ALL subtypes is relevant to the field, which has long speculated about a common underlying mechanistic association between infection and ALL. Despite the novel insights provided, however, the study has significant weaknesses in its current form, most notably the lack of appropriate control or comparator groups in the reported experiments.

We greatly appreciate this kind appraisal by Reviewer 1, and the thoughtful comments below which have significantly strengthened the revised manuscript. We have carefully addressed all comments in the revised manuscript, as detailed below.

Major concerns:

1. *The effect of antibiotics in the additional models (Figure 1: BCR-ABL, Lmo2, and ETV6-RUNX1) is compared to that reported previously for Pax5^{+/-} mice; critically, no concurrently obtained results for the Pax5 model are presented. Given the significant impact reported for the poorly understood variables (i.e. SPF vs conventional housing) on disease progression, the potential that the conditions during the historical Pax5 study have changed cannot be dismissed. Without a demonstration that Abx-treated Pax5 mice developed leukemia in experiments performed in the same time frame as the additional models, the statement that it behaves differently is not sufficiently supported.*

We fully agree with the reviewer on this point. In fact, we had not included this information in

our original submission because, in the past, some reviewers complained that this was tantamount to presenting already published data. Now, following this reviewer advice, the revised version includes the control cohort of Abx-treated Pax5+/- mice that were aged during the same time frame in the SPF facility (**new Figure 1C**), confirming that Abx-treated Pax5+/- mice developed B-ALL, and reproducing our previously described results (Vicente-Dueñas et al. Blood 2020, 36(18):2003-2017).

2. Similarly, in the early vs late exposure studies (Figure 2), survival of Pax5 mice maintained under SPF conditions over the same time period should be shown to confirm that the model retains the same need for infection and penetrance that was previously reported. These controls are essential while the precise nature of the environmental change driving leukemogenesis is unknown and thus cannot be shown to be consistent between experiments performed at different times.

We thank the reviewer for this suggestion. Since 2015, we have been keeping in our facilities both WT and Pax5+/- mice in specific pathogen free (SPF) and conventional facilities (in the latter, they are exposed to infections) to be sure that environmental factors remain stable. Pax5 heterozygous mice do not spontaneously develop B-ALL in SPF (J Exp Med 2011, 208: 1135-49; Nat Genetics 2014, 46: 618-623, Cancer Discov. 2015 Dec;5(12):1328-43; Nat Commun. 2019 Dec 5;10(1):5563). To confirm this fact in the case of the present study, and following the reviewer's advice, we now present a new cohort of Pax5+/- mice born and kept in the SPF environment for 2 years in parallel with the experimental groups mentioned in the present study. These mice are presented in **new Figure 2B**. Thus, the revised manuscript version now includes the control cohort of Pax5+/- mice that were aged for the same time in the SPF facility, confirming that the exposure to infection is the causal factor.

3. While it is true that Myd88+/- Pax5 mice show higher leukemia penetrance (Figure 3), it is an extremely late effect, only becoming apparent at a time when leukemia onset in Pax5+/- mice appears to have ceased. Given this timing, it is essential to show survival curves for Myd88+/- mice to show they do not develop leukemia after 20 months (which is not an unreasonable prediction given the inflammatory signature reported)? The need for this control is compounded by the claim (without supporting disease progression kinetics data) that leukemia occurs without "infection" in the double transgenics.

This is indeed an important remark. Accordingly, in the revised version we now show survival curves for Myd88+/- mice under exposure to infection together with the survival curves for both Myd88+/- and Myd88+/-; Pax5+/- mice kept in SPF (**new Figure 3B**)

4. The reduction in Myd88 expression induced by antibiotic treatment of Pax5 mice is a central to the conclusions of this study (Figure 3). It is surprising, therefore, that the same analysis was not performed in ETV6-RUNX1 mice, which is also develops "infection-driven" ALL but does not show the same influence of antibiotic treatment. Such data would greatly inform the interpretation of the presented results and help to clarify the generality of such effects.

We thank the reviewer for this constructive criticism. Accordingly, in the revised version we have now included a **new Figure 3A** showing that antibiotic treatment of ETV6-RUNX1 mice induces a reduction in Myd88 expression in precursor B-cells but the Myd88 expression levels in ETV6-RUNX1 precursor B-cells after antibiotic treatment are similar to the Myd88 expression levels in untreated precursor B-cells of both WT and pax5+/- mice. These findings are in agreement with the fact that antibiotic treatment does not promote B-ALL development in the Sca1-ETV6-RUNX1 mice.

5. While the impact of Poly(I:C) on Pax5 leukemia development is interesting, the nature of the effect is not clear. Evaluation of acute effects of this treatment on the preleukemic cell population would be very informative. More importantly, the lack of any data on the effect of Myd88-mediated TLR signalling on Pax5+/- leukemia progression (in the presence and absence of infection exposure) is a significant omission, given the critical position in leukemogenesis proposed for this axis.

Following the reviewer's insightful suggestion, in the revised version we have now evaluated the immediate effects of Poly(I:C) treatment on the preleukemic cell population. Previous papers have already shown the effect of Myd88-mediated TLR signalling on ETV6-RUNX1-carrying pre-B cells (LPS activation; Swaminathan et al., Nat Immunol 2015), on B cell lymphoma (by Plasmodium falciparum; Robbani et al., Cell 2015) and on Pax5+/- precursor B-cells, as a driver of clonal evolution towards B-ALL (LPS activation; Nat Commun. 2019 Dec 5;10(1):5563). In the revised version, we have now showed that the preleukemic compartment is reduced in Pax5+/- mice as an immediate effect of Poly(I:C) treatment (**new Supplementary Figure S26**), and we further indicate that the effect of Myd88-mediated TLR signalling on Pax5+/- leukemia progression is independent of infection exposure, as Myd88+/-;Pax5+/- mice develop leukemia even when they are kept in an SPF facility (**new Supplementary Figure S9**).

6. The study is presented in the context of the hypothesized causal role of infection on pediatric B-ALL, an etiology in which age of infection is thought to be important. However, leukemogenesis in the Pax5 model occurs relatively late (very late in the context of the higher penetrance in Pax5/Myd88 mice). This aspect of the model needs to be directly discussed.

This is an excellent question, that we have addressed in our recent review (Nat Rev Immunol. 2021 Sep;21(9):570-581.) but we do not yet have an explanation for this interesting finding. We believe that the time to leukemogenesis is, to a large extent, related to the species-specific differences in the dependence of early B cell progenitors to IL7. Murine early B cell progenitors are strongly dependent on IL7 signaling and, when IL7 is removed from cell cultures, these cells die. IL7, when bound by its receptor activates Jak3 and phosphorylates STAT5. We consider that this is the cause why, under the selection pressure of exposure to a common infectious environment, murine B cell progenitors need more time to acquire Jak3 mutations in comparison with their human counterparts. This phenomenon has, to our knowledge, not been observed in humans yet, probably because human B cell progenitors are to a large extent independent of IL7 (Ghia et al. Immunol Today. 1998 Oct;19(10):480-5). We have added a comment to explain this more clearly on pages 14-15 of the revised version of the manuscript.

Minor points:

"SPF (i.e., germ-free)" is not an accurate statement.

This is indeed an error and has been now corrected removing the "(i.e. germ-free)" analogy to SPF conditions in this sentence.(page 9).

Supplementary Fig. S8B: The statement "... decrease in Myd88 expression was also observed in Sca1- ETV6-RUNX1 mice where the appearance of B-ALL is also triggered by infection exposure" does not appear to be supported by the data shown on the figure. Stats should be provided to show significance.

In the revised version this sentence has been replaced by: “The antibiotic treatment of ETV6-RUNX1 mice also induces a reduction in Myd88 expression in precursor B-cells but the Myd88 expression levels in ETV6-RUNX1 precursor B-cells after antibiotic treatment are similar to the Myd88 expression levels in untreated precursor B-cells of both WT and pax5+/- mice (**Figure 3A**), in agreement with the fact that antibiotic treatment does not promote B-ALL development in the Sca1-ETV6-RUNX1 mice)”

S14B: The description of GSEA results for Pax5+/-, Pax5+/-;Myd88+/-, and human leukemia cells is either contradictory between the main text and the figure legend or needs to be rephrased for better clarity.

We apologize for this mistake. We have now corrected this in the revised version of the manuscript, where we present the microarray and subsequent GSEA analysis in a different format, following Reviewer #3’s advice (page 11 and new Supplementary Figures S13-S15).

Reviewer #2 (Remarks to the Author):

The authors utilized the B-ALL development of genetically modified mouse models and examined the effect of infection on B-ALL development. They used several mouse models, such as Sca1-BCL-ABL, Sca1-Lmo2, and Sca1-ETV6-RUNX1. Although infection affects B-ALL development in Pax5+/- model, the infection did not affect B-ALL development in the three mice models used in this study. They tested early- and delay-exposure models and found no difference between the two models. Since MyD88 plays a key role in innate immunity, they measured MyD88 expression levels and examined the effect of MyD88 +/- genetic background on B-ALL development. They also tested MyD88-independent stimulation using poly I:C. Unfortunately, this manuscript is not well organized, and several experiments are not performed appropriately. Thus, I am afraid that they failed to provide sufficient data to support their conclusions. Additionally, the underlying mechanism of the relationship between innate immunity and B-ALL development remains unclear. Considering these weaknesses in this manuscript, This manuscript is too preliminary. Specific comments are described below.

We thank Reviewer 2 for the thorough review and helpful comments and suggestions for improving our manuscript. Reviewer 2 indicates, quite rightly, some weak points. Now we have carefully addressed all of these comments in the revised manuscript, as detailed below.

Comment 1:

In Fig 1C, the authors showed that the Sca1-Lmo2 and Sca1-BCR-ABLp190 models developed B-ALL in a gut microbiota-independent manner, but it is unclear whether the Abx-treatment was sufficient. Pax5+/- should be included in Fig 1C experiment as a control. Alternatively, they should provide data showing that Abx-treatment was sufficient to exclude gut microbiota. In addition, statistical analysis should be performed to clarify whether the difference in leukemia-specific survival between Abx-treated and untreated Sca1-Lmo2 mice was significant or not.

We thank the reviewer for this constructive criticism. Accordingly, in the revised version we have included a control cohort of Abx-treated Pax5+/- mice that were aged in the same time frame in the SPF facility (**new Figure 1C**), confirming that Abx-treated Pax5+/- mice developed B-ALL as previously described (Vicente-Dueñas et al. Blood 2020, 36(18):2003-2017). Although we had this information, we had not included it in our original submission

because, in the past, some reviewers complained that this was presenting already published data. The statistical analysis confirmed that the difference in leukemia-specific survival between Abx-treated and untreated Sca1-Lmo2 mice was not significant (now included in Figure 1 legend).

Comment 2:

In Fig 2, the authors tested the early- and delayed-exposure groups. I am afraid that this experiment is a little confusing. There are many microorganisms even in SPF condition, and thus it is different from the pathogen-free condition. Moreover, Fig 2B lacks several important controls. In the left panel, Pax5+/- no exposure (pathogen-free condition) should be included as a control.

We thank the reviewer for this suggestion. Since 2015, we have been keeping in our facilities both WT and Pax5+/- mice both in SPF and exposed to infections to be sure that environmental factors remain stable. Pax5+/- mice do not spontaneously develop B-ALL in SPF (J Exp Med 2011, 208: 1135-49; Nat Genetics 2014, 46: 618-623, Cancer Discov. 2015 Dec;5(12):1328-43; Nat Commun. 2019 Dec 5;10(1):5563). We again corroborated these findings when Pax5+/- mice were housed under specific pathogen free (SPF) conditions in parallel with the experimental groups mentioned in the present study. Following the reviewer's suggestion, we now present this new cohort of Pax5+/- mice born and kept in the SPF environment for 2 years. These mice are presented in **new Figure 1C**. Thus, the version now includes the control cohort of Pax5+/- mice that were aged for the same time in the SPF facility, confirming that the exposure to infection is the causal factor. In addition we had included the microbiological status of our facility (Table S4).

In addition, the authors should investigate several mice and perform statistical analyses in Fig 2C.

The diagnosis of B-ALL in each individual mouse is systematically based on the presence of blast cells in BM and hematopoietic tissues. These blast cells are never present in leukemia-free mice. Figure 2C just illustrates the presence of blast cells in two cases.

Comment 3:

In Fig. 3, the authors investigate the expression of MyD88 and its relationship with B-ALL-specific survival. But the authors' explanation seems to be confusing. For instance, if the authors want to see the effect of MyD88, the experiments should be performed in MyD88 knockout background, not heterozygote.

We thank the reviewer for this constructive comment. We used MyD88+/- mice because in preleukemic cells, we observed downregulation of Myd88, not lack of Myd88 expression. In fact, the level of Myd88 in Pax5+/-;Myd88+/- leukemic cells is similar to the level of Myd88 in antibiotic-treated Pax5+/- leukemic B cells (Figure 3A). Nevertheless, the experiments could not be performed in MyD88 knockout background due to the lethality of Myd88-deficient mice because of their increased susceptibility to bacterial and viral infection caused by their immune system abnormalities (Immunity, Vol. 9, 143–150, July, 1998).

Comment 4:

In Fig 5, the authors used poly I:C as a Myd88-independent ligand. It has been reported that poly I:C can be recognized by TLR3, RIG-I, and MDA5 in vivo. To test the effect, the authors should use TLR3, RIG-I, or MDA5 KO mice.

We appreciate Reviewer #2's insightful critique regarding the use of poly I:C on the ability of non-Myd88 TLR signaling to delay leukemia onset in the Pax5 model. Our hypothesis was that, since Myd88-dependent and Myd88-independent pathways share the final steps, non-Myd88-mediated TLR signaling might affect the leukemia onset in the Pax5 model. The fact that the aim was achieved suggests, as the reviewer indicates, that a similar effect to that seen with My88^{+/-};Pax5^{+/-} mice could be observed with TLR3-deficient mice. However, TLR3 downregulation is not a natural occurring event observed in Pax5-het preleukemic cells under immune stress and, therefore, we think that these experiments are beyond the scope of this study.

Alternatively, MyD88-dependent ligands, such as LPS and peptidoglycan should be compared with poly I:C.

Previous papers have already shown the effect of Myd88-mediated TLR signalling on ETV6-RUNX1-carrying pre-B cells (LPS activation; Swaminathan et al., Nat Immunol 2015), on B cell lymphoma (by Plasmodium falciparum; Robbiani et al., Cell 2015) and on pax5-het precursor B-cells, as a driver of clonal evolution towards B-ALL (LPS activation; Nat Commun. 2019 Dec 5;10(1):5563). The effect of Myd88-mediated TLR signalling on Pax5^{+/-} leukemia progression is independent of infection exposure as My88^{+/-};Pax5^{+/-} mice develop leukemia even when the mice are kept in an SPF facility (**new Supplementary Figure S9**).

Comment 5:

Although they focused on innate immunity, the underlying mechanism observed in Figures 3 and 5 remains unclear. The authors should examine the innate immune responses, such as cytokine expression and accumulation of myeloid cells.

We thank the reviewer for this constructive comment. Following this advice, we have now included the analysis of myeloid cells in My88^{+/-};Pax5^{+/-} mice in the **new Supplementary Figure S18**. The results show that My88^{+/-};Pax5^{+/-} mice have a similar myeloid compartment to WT, Myd88^{+/-} and Pax5^{+/-} mice. In addition, we have measured concentrations of inflammatory cytokines (IL-2, IL-4, IL-6, IL-10, IL-17A, TNF alpha and IFN gamma) in serum from WT and Pax5^{+/-} and My88^{+/-};Pax5^{+/-} mice (**new Supplementary Figure S22**). After performing the Kolmogorov-Smirnov normality test, it appears that the samples do not conform to a normal distribution, so the statistical tests to be applied will be non-parametric. The Kruskal Wallis test comes out for TNFα p<0.001 and IL4 p=0,044. These new results show an increase in inflammatory cytokines in the serum of Myd88^{+/-};Pax5^{+/-} mice in agreement with the transcriptional profiles of Pax5^{+/-};Myd88^{+/-} preleukemic precursor B-cells.

As a result of poly(I:C) treatment we showed that observed a similar increase in inflammatory cytokines and CD8⁺ cells in peripheral blood in both WT and Pax5^{+/-} mice (**Figure 5B and Supplementary Fig. S23-S24**). Now, following the reviewer's advice, we also examined the accumulation of myeloid cells, showing that poly(I:C) treatment did not induce an accumulation of myeloid cells in both WT and Pax5^{+/-} mice (**new Supplementary Figure S27**). In addition, we have measured concentrations of new inflammatory cytokines using the the Cytometric Bead Array (CBA) immunoassay systems (BD Biosciences), which assesses simultaneously IL-2, IL-4, IL-6, IL-10, IL-17A, TNF alpha and IFN gamma in serum from WT and Pax5^{+/-} mice exposed to poly(I:C) (**new Supplementary Figure S25**). After performing the Kolmogorov-Smirnov normality test, it appears that the samples do not conform to a normal distribution, so the statistical tests to be applied will be non-parametric (the Kruskal-Wallis test). These new data further support that poly(I:C) administration was having a similar immune-activating effect in WT and Pax5^{+/-} mice.

Minor comment 1:

The genetic constructs are unclear. For instance, the construct of Sca1-ETV6-RUNX1 should be explained sufficiently. In addition, the Sca1 construct is frequently used in this manuscript, but it is not so familiar.

We appreciate this criticism and have attempted to better explain the construct of Sca1-ETV6-RUNX1 and the Sca1 construct itself in Method section as follows: “*The Sca1-ETV6-RUNX1 construct was generated by placing the human ETV6-RUNX1 cDNA under the control of the stem-cell-specific Sca1 promoter, as has been previously described (Rodriguez-Hernandez et al., 2017) ”* (page 17-18).

Minor comment 2:

There is a little long introduction in the first paragraph of the result section. Most of those sentences should be included in the introduction.

In the revised manuscript following Reviewer’s advice, we have moved the first paragraph of the result section to the introduction (pages 4-5).

Reviewer #3 (Remarks to the Author):

In this study, the authors Isidro-Hernández et al try to decipher the mechanistic role of Myd88 loss in promoting the onset of leukemia in Pax5+/- mice. This is an important model to study since the role of infection in the development of leukemia has been known for some time. But the exact pathogenesis is yet to be well laid out.

Overall, the paper is quite interesting and provides us with some novel insights about leukemogenesis in a Pax5+/- model. The immune upregulation after Myd88 knockdown appears quite significant and the poly I:C administration does slow down the onset of leukemia considerably. However, the paper does suffer from numerous concerns which are listed below.

We thank Reviewer 3 for the thorough review of our manuscript and very positive comments. We have carefully addressed all questions in the revised manuscript, as detailed below.

Comments:

Figure 1A: The authors show that the microbiome of all the genotypes is different on the PCA plot. However, the explained variance is quite small in all the 3 axes. Can the authors also check for more principal components (PC4, PC5, etc.) to check whether they can explain the variance further?

In the revised manuscript, following the Reviewer’s advice, we have now checked for more principal components (PC4 and PC5) that are now included in **new Figure 1A**. We have used the PCA plot shown in Figure 1A just as a representation to visualize the microbiome beta diversity distances between all samples in the study. To test whether there are differences between samples we used Pairwise Permanova tests represented in Figure 1B where the p-values for each comparison are given.

Figure 1C: Can add the significance values between the pairs or NS to show that there is no difference between the antibiotic treated and non-treated groups in leukemia development. Even though there was no difference in survival, was there any difference in the histopathology of the tumors which developed? This may provide us with some clue about the differences in the microbiome contributing to phenotypic differences in the tumors.

Figure 1 demonstrates that disrupting the gut microbiome does not cause any change in leukemogenesis onset in different models. However, this is not investigated further. So, this figure is a little disjointed from the rest of the work.

We appreciate this criticism and have attempted to better explain this aspect in the revised version. The statistical analysis is now included, and confirmed that the difference in leukemia-specific survival between Abx-treated and untreated mice was not significant, and the disease characteristics from both immunophenotype and histological points of view are identical (page 7). The aim of these initial experiments was to explore a possible link between the variables suspected to influence pediatric ALL onset. The results indicate that the drivers of leukemia progression may differ between ALL subtypes, therefore answering a question that has been subjected to much speculation for many years, about the existence of a common underlying mechanistic association between infection and different ALL initiating lesions (pages 7).

Figure S1: The text is not legible.

We apologize for this mistake and have now corrected in the revised version of the manuscript by modifying the Taxa Bar plot of the Microbial signatures by using 4 taxonomic levels instead of 5 and by amplifying the size of the legend. We have also deleted panels B to D from Supplementary Figure S1 as the text is not legible and the data is already included in Supplementary Tables S1-S3. We have also modified the legend of those tables to be more informative.

Figure 2: The survival curves compare WT and Pax5+/- mice after early exposure to infections. Was Pax5+/- grown in SPF also used as a control? That would have been an ideal control to check for the phenotype which was intrinsic to the loss of one Pax5 allele with no infection exposure.

We thank the reviewer for this suggestion. Since 2015, we have been keeping in our facilities both WT and Pax5+/- mice both in SPF and conventional facilities (in the latter they exposed to infections) to be sure that environmental factors remain stable. Pax5 heterozygous mice do not spontaneously develop B-ALL in SPF (J Exp Med 2011, 208: 1135-49; Nat Genetics 2014, 46: 618-623, Cancer Discov. 2015 Dec;5(12):1328-43; Nat Commun. 2019 Dec 5;10(1):5563). We again corroborated these findings when Pax5 heterozygous mice were housed under specific pathogen free (SPF) conditions in parallel with the experimental groups mentioned in the present study. Therefore, following the reviewer's suggestion, we now present a new cohort of Pax5+/- mice born and kept in the SPF environment for 2 years. These mice are presented in **new Figure 2B**. Thus, the version now includes the control cohort of Pax5+/- mice that were aged for the same time in the SPF facility, confirming that the exposure to infection is the causal factor.

Have the authors checked for the flowcytometry based immunophenotype of the hematopoietic tissues of Pax5+/- mice grow in SPF conditions?

We already examined hematopoietic tissues of Pax5+/- mice grow in SPF conditions by flow cytometry. The results were published in Cancer Discov. 2015 Dec;5(12):1328-43.

The flowcytometry data shows an expansion of CD19-IgM-B220+ cells in Pax5+/- mice after infection exposure and supplementary figure S4 shows infiltrating blast cells in various organs. A higher magnification of the histopathology should be provided with the infiltrating malignant cells marked out.

Following reviewer's advice, in the revised version we now include a higher magnification of the histopathology with the infiltrating malignant cells marked out (**new Supplementary Figure S4**).

Did the authors find leukemic cells in the peripheral smear/bone smear examination? Those images can also be included.

Unfortunately, we do not include PB smears or BM smears in our mouse leukemia analysis because the B-ALL diagnosis is based on flow cytometry findings.

Figures S4 and S6 would be served better by including a Pax5^{+/-} control grown in SPF condition. There appears to be a difference between WT (early vs late exposure) and WT vs Pax5^{+/-} for number of B220⁺ positive cells. Instead of ANOVA, the authors can use t-tests/Mann-Whitney to compare between two groups for significance.

Following the reviewer's advice, in the revised version we now include a Pax5^{+/-} control grown in SPF conditions as a reference for histology in new supplementary Figure S4. Similarly, we have replaced the ANOVA test with the Mann-Whitney test used to compare between groups for significance in Figure S6. However, we have not included in Figure S6 the data related to the percentage of B-cells in PB from Pax5^{+/-} mice grown in SPF condition as these data are already published in Figure 3B of the Cancer Discov. 2015 Dec;5(12):1328-43-PMID: 26408659. In the current manuscript, we are comparing the effect of early versus delayed exposure to infections in peripheral blood B cells.

Supplementary Figure S7: How was this experiment done? What was the source of the tumor tissue? Were cells sorted or purified in any way? What was the source of germline control? A detailed methodology would help in understanding the results better. As we mention in the legend of Supplementary Figure S7, tumor DNA was derived from the bone marrow where the percentage of blasts cells was higher than 80% in each B-ALL and germline control DNA was obtained from the tail when the mice were 4-weeks-old. This point has been further clarified in page 22 in Methods section in the revised version of the manuscript.

Supplementary Figures S8A-B: The error bars are quite high for most of the graphs and there does not seem to be any significant decrease in Myd88 expression in most conditions. Can the authors clarify the exact significance levels and the reason for the variation? The reason for the variation is that changes in Myd88 are only present in a small percentage of Pax5^{+/-} preleukemic mice.

Supplementary Figure S9: This is not a valid comparison since healthy marrow would have >70% of myeloid cells and their comparison with leukemic cells does not make much sense.

We thank the reviewer for this constructive comment and we fully agree with it. We are not aware of any such information on human B preleukemic cells. Indeed, most of the patients of which we are aware have already developed leukemia. To the best of our knowledge, bone marrow aspirates are not routinely performed in individuals harbouring pathogenic germline PAX5 variants or ETV6-RUNX1 fusion transcripts who have not yet developed leukemia (according to the available literature, it is not currently recommended that these patients undergo bone marrow analysis – please see: <https://pubmed.ncbi.nlm.nih.gov/28572263/>). For this reason, we used healthy marrow as a control. In the revised version, and following reviewer's advice, we have removed the Supplementary Figure S9.

Figure 3: Were pre-pro B cells sorted from all samples? If so, what markers and methodology were used? The authors should clarify what they mean by 'pre-leukemic'? In fact, the text mentions precursor B-cells while the figure says pre-pro B cells.

As we mentioned in Methods: "Expression of *Myd88* was analyzed by Q-PCR in preleukemic samples (sorted BM precursor B (B220^{low} IgM⁻) cells of wild-type (n=3), Pax5^{+/-} (n=3) and Pax5^{+/-} mice treated with Abx for 8 weeks (n=3)) as well as in leukemic samples". In the revised version, the nomenclature of preleukemic cells has been fixed.

Myd88 is a critical component in innate immune signaling. Knockout of the same is associated with numerous abnormalities in the immune cell phenotyping. The authors should have used a Myd88^{+/-} Pax5 WT control which would have been ideal instead of a WT control for the flow cytometry and histopathology.

Since we had already collected these data over the course of the experiments, following reviewer's advice, in the revised version we now add a Myd88^{+/-} mouse control for the flow cytometry and histopathology (**revised Figure 3 and Supplementary Figure 10**).

The difference in survival between Pax5^{+/-} and the Pax5^{+/-}Myd88^{+/-} is not too different upto ~18 months and shows much after that. Am not sure how significant this would be for translational value.

The difference in survival between Pax5^{+/-} and the Pax5^{+/-}Myd88^{+/-} is highly significant (p-value=0.0092) as indicated in Figure 3 legend.

The authors also mention that under SPF conditions, Pax5^{+/-} mice don't develop leukemias while the Pax5^{+/-} Myd88^{+/-} mice do. I could not find the data for the same (immunophenotyping, histopathology, etc.). Can the authors add the same since it is a significant result?

Thank you very much for this very important remark. Accordingly, in the revised version we now show survival curves for Pax5^{+/-}Myd88^{+/-} mice under exposure to infection close to survival curves for both Pax5^{+/-} and wild type mice kept in SPF (**new Supplementary Figure S9**). In addition, we include two panels (B and C) within the figure showing immunophenotyping and histopathology of B-ALL in Pax5^{+/-} Myd88^{+/-} mice kept in SPF.

Figure 4: From the analysis in Figure 4A, were the authors able to identify which subtype of B-ALL did the tumors resemble transcriptomically? Some of the pathways identified in Supplementary Figure S13 (EMT, Myogenesis) may not be practically significant. The heatmap shows WT pre- and pro-B cells. What are these cells? The methodology is inadequate at many places and the terminology for the controls is used very interchangeably which makes it difficult to comprehend. A better experiment would be the comparison of the transcriptome of leukemic cells from Pax5^{+/-} Myd88 WT, Pax5 WT Myd88^{+/-} and Pax5^{+/-} Myd88^{+/-} mice.

In fact, the microarray and subsequent GSEA analysis can be more detailed (maybe in the supplementary data) mentioning the packages used to analyze, number of genes differentially expressed between all groups followed by the list of pathways significantly up/downregulated in the GSEA (using Hallmark/KEGG/GO pathways) with enrichment ratios. The present figures appear inadequate to explain the data from the experiment.

We appreciate the reviewer's constructive comments. We have now corrected this in the revised version of the manuscript, where we present the microarray and subsequent GSEA analysis in a different format, following the reviewer's advice as follows:

We have now indicated the B-ALL subtype that the murine leukemias resemble to transcriptomically in Figure 4A. To be more precise, murine leukemias resemble BCR-ABL

and ETV6-RUNX1 human leukemias and the list of genes are now included in **new Supplemental Figures S13B-D RELATED TO MAIN FIGURE 4A.**

We have also now explained in the figure legend of Supplementary Figure S13 what are WT pre- and pro-B cells as follows: **“proB and preB cells (bone marrow B220^{low} IgM cells) from control wild type (WT) mice...”**

The comparison of the transcriptome of leukemic cells from Pax5^{+/-} Myd88 WT and Pax5^{+/-} Myd88^{+/-} mice is shown in Supplementary Fig. S15. Pax5 WT Myd88^{+/-} mice were not included in the comparison because they do not develop leukemia as we now indicate in page 10 of the revised manuscript.

Following reviewer's advice, the GSEA analysis has been completed by including the list of pathways significantly up/downregulated in the GSEA (using Hallmark/KEGG/GO pathways) in the revised version of Supplementary Fig. S14, Supplementary Fig. S15B-D, Supplementary Fig. S20 A-C and in Supplementary Fig. S21. Enrichment scores are now included in all of the GSEA analyses.

The GSEA analysis is explained in detail in the Methods section as follows:

“Samples were analyzed using Applied Biosystems Transcriptome Analysis Console (TAC) software v4.0.1 with the following parameters: (i) Analysis Version: version 1, (ii) Summarization Method: Gene Level – RMA, (iii) Genome Version: mm10 (Mus musculus) and (iv) Annotation: MoGene-1_0-st-v1.na36.mm10.transcript.csv. Differentially expressed genes were evaluated by specific statistical criteria (p-value <0.05 and absolute fold change >2). Gene Set Enrichment Analyses were performed using GSEAv4.3.2⁵⁵⁻⁵⁷, hallmark collection of gene sets⁵⁵, canonical pathways gene sets derived from the KEGG pathway database, gene sets derived from the GO Biological Process ontology, murine B cell developmental stages gene sets⁵⁸, and Pax5-regulated gene sets^{59,60} were analyzed. Gene expression signatures that are specifically upregulated or downregulated in human B-ALL²⁸⁻³¹ were also tested for enrichment within tumor specimens using GSEA.”

Figure 4B: The authors mention that the mutations observed in Pax5^{+/-} tumors and Pax5^{+/-} Myd88^{+/-} tumors are not different. Did the authors perform any statistical analysis for the same?

The sentence mentioning that the mutations observed in Pax5^{+/-} tumors and Pax5^{+/-} Myd88^{+/-} tumors are not different refers to the fact that the same genes are mutated in both cases. This point has been clarified in the revised version (page 12).

Figure 4C and Supplementary Figure S19: The interpretations in the results section are too speculative. It is clear from Figure 4C that immune signaling pathways are upregulated after Myd88 knockdown in a Pax5^{+/-} background (Healthy Pax5^{+/-} Myd88^{+/-} vs. healthy Pax5^{+/-} Myd88 WT). However, the comparison in Figure S19 is not correct: WT vs. Pax5^{+/-} Myd88^{-/-}. There are too many variables in the KO and making conclusions that ‘this inflammatory transcriptional profile was not present under complete loss of Myd88 in Pax5^{+/-} pro B cells, therefore being associated with intermediate, but not zero, levels of Myd88 expression’ does not seem correct. The correct comparison would be Healthy Pax5^{+/-} Myd88 WT vs. healthy Pax5^{+/-} Myd88^{+/-} vs. healthy Pax5^{+/-} Myd88^{-/-} for meaningful conclusions.

We thank the reviewer for this constructive comment. In the revised version we have omitted the comparison with Pax5^{+/-} Myd88^{-/-}.

Figure 5: Poly I:C treatment does not seem to cause much difference in CD8+ cell numbers as well as CD4+ and B-cells. In fact, the graphs intersect for the WT CD8+ cells. Can the authors add statistics and strengthen their statement?

Unpaired t-test confirmed that the percentage of CD8+ T cells in the peripheral blood increases after the poly(I:C) treatment in the Pax5+/- mice (p-value=0.0087). As described, poly(I:C) treatment specifically increases CD8+ T cells. Likewise, the levels of CD4+ T cells and B cells do not increase after the treatment (CD4+ T cells: Pax5+/- mice treated with poly(I:C) vs untreated Pax5+/- mice p-value=0.3485 and WT mice treated with poly(I:C) vs untreated WT mice p-value=0.4782; B cells: Pax5+/- mice treated with poly(I:C) vs untreated Pax5+/- mice p-value=0.4177 and WT mice treated with poly(I:C) vs untreated WT mice p-value=0.2707). This point has been clarified in the revised version (page 14).

The legend has too much methodology and results. Can shorten it considerably.
In the revised version we have shortened the legend.

There is a clear difference in onset of B-ALL between the untreated and poly I:C treated groups. Was there any difference in the histopathology?

There was not any difference in the histopathology and this point has been added in the revised version (**new Supplementary Figure S29** and page 14)

Supplementary Fig. S21: The figures are not legible. Paired t-tests would be appropriate in this case.

Supplementary Fig. S21 (Supplementary Fig. S24 in the revised version) has been remade and significance estimated using Paired t-tests according to the reviewer's recommendations.

Supplementary Figs S22-23: Figures from Control Pax5+/- mice can be added for comparison.

Following reviewer's advice, in the revised version we have included control Pax5+/- mice (Supplementary Fig. S28 in the revised version).

Minor comment: The manuscript would benefit from some spelling and grammar checks to fix multiple errors.

Spelling and grammar checks have been done in the revised version

REVIEWER COMMENTS

Reviewer #1 (Remarks to the Author):

I appreciate that the authors have made significant changes to this manuscript. Importantly, these include the addition of Pax5 model groups in the experiments presented in Figures 1 and 2. These data, obtained over the same time period as the other reported experiments, are essential for the validation of the foundational findings of the manuscript. In my opinion, the manuscript is strengthened by the inclusion of these comparative data and remains an intriguing one that provides evidence of a new immune activity in driving B-ALL progression.

However, the lack of mechanistic insight provided by the current experiments continues to reduce enthusiasm for the manuscript in its current form. While results from additional experiments have been provided, these do not wholly address the need for more detailed explanations for the observed outcomes. Specifically:

1) The lack of data from the use of defined infections or danger signals remains the biggest weakness in this study. In the resubmission cover letter, the authors refer to previous papers showing effects of MyD88 signaling on B-ALL; however, those BCP ALL studies have all involved in vitro exposure to TLR ligands i.e have not incorporated the impact of systemic TLR-mediated immune stimulation on in vivo leukemogenesis. Such models do not shed much light on the balance of mechanisms at play in a Pax5^{+/-} mouse. If, as the authors contend, Myd88 insufficiency mimics the role of infection as a driver of young onset B-ALL, then the exposure of SPF-housed Pax5^{+/-} mice to infectious agents or TLR ligands should generate leukemia. Critically, such experiments would begin to reduce the number of unknown variables in this model, such as the nature of infection, timing of exposure, pathway specificity, age-related immune competence. The need for such insights is demonstrated by the failure of MyD88^{+/-} Pax5^{+/-} double transgenics to show earlier onset of disease than Pax5^{+/-} mice. While an explanation for the late onset is proposed, the fact that all additional deaths occur after 15 months of age in mice with constitutional immune dysregulation raises concerns this is not recapitulating childhood B-ALL leukemogenesis but rather reflecting activities exerted by an old immune system.

2) In contrast to the lack of Myd88-dependent TLR ligands, the authors have added data regarding the acute effects of Poly(I:C) treatment on the “preleukemic” BCP cell population (confusingly, the text on p13 states that the treatment did deplete preleukemic cells, while the figure legend states that it did not. Similar contradictory statements are made for the myeloid enrichment after treatment). This is likely just a typo issue, but it draws attention to the lack of clarity as to what constitutes a preleukemic cell and whether this assay has the resolution to detect the depletion of genuine pre-malignant BCP cells. The increased latency is attributed to “early Myd88-independent TLR ligation”, but in the absence of the paired experiment with a Myd88-dependent ligand, it is impossible to determine how Poly(I:C) extends the disease latency or whether it really induces an outcome distinct from Myd88-dependent TLRs.

3) While the inclusion of data from the ETV6-RUNX1 model on Myd88 expression after antibiotic treatment is appreciated, the interpretation of the result is not convincing. It appears from Figure 3A that, although the mean is similar to that of untreated Pax5 mice, half of the ETV6-RUNX1 mice reduce Myd88 expression to levels seen in Pax5^{+/-} Abx mice and Pax5 leukemia. The inability of Abx-treatment to drive leukemia progression in the ETV6-RUNX1 model, therefore, does not appear to be due to a failure to down-regulate

Myd88 expression. While the different leukemia models may have very different reliance on Myd88 signaling, it highlights the need to identify downstream mechanism(s) to provide possible explanations for such differences.

Overall, the authors have addressed weaknesses in the initial submission, especially by providing necessary controls. However, in the absence of a deeper investigation of the mechanism underlying Myd88's influence on leukemia progression (eg whether it is Myd88 signaling within preleukemic BCP or the broader immune system that is the key driver), the study does not provide sufficient insight to merit publication in Nature Communications.

Reviewer #2 (Remarks to the Author):

In this revision, the authors have improved the manuscript throughout. They have addressed all of my previous concerns appropriately.

Reviewer #3 (Remarks to the Author):

The authors have made a lot of effort to improve this manuscript which is quite commendable. Especially the addition of more control groups in the mouse experiments is very heartening since it contributes a lot to scientific rigor. Most of the major queries have now been resolved. However, there are a few more minor comments which would help in polishing the manuscript further.

Figure 1B: The X-axis should ideally begin from 0. This makes the data look spuriously inflated.

Figure S3: The legend mentions about the thymus being infiltrated which is not seen in the flow cytometry data; The data about C-kit and CD25 positivity mentioned in the legend is not seen in the graphs.

Figure S6: The early exposure groups (including the WT) show a significant decrease in PB B-cells. The authors should speculate a bit on this phenomenon; esp. the decrease in the WT mice.

Figure S8: This data is still too haphazard to me with large error bars in most of the samples and genes. The authors can try to plot the data using a scatter plot or plot only the relative expression instead of fold changes to make it clearer.

Figure 3 legend: 'Development' instead of 'emergency'

Figures S14-S15: Did the authors find any upregulated signature for B-ALL subtype when they compared the transcriptome of leukemic Pax5+/- Myd88WT with the Pax5+/- Myd88+/- cells? Since the authors compared the human transcriptomic signatures with mouse transcriptome, the methodology can be elaborated for the use of readers, maybe in the supplementary information. Also, the figures showing the NES and FDR can be represented as bubble plots which can also show enriched gene count along with ES and FDR/p-value (S14-15. S20-21).

Figure S19: Can add a composite Venn diagram with DE gene numbers to get an idea of the degree of overlap between groups.

Figure S22, S24 and S25: Are not legible. Can be redrawn clearly and text to be added in English.

Figure S26: The authors can try to plot absolute numbers of cells instead of the percentages if they have the bone marrow cell counts. This may provide better translational value than the percentages alone.

On a related note, there definitely appears to be some B-cell developmental changes in the bone marrow. The authors can try to study Hardy Fractions in the bone marrow of the Pax5^{+/-} exposed to infection/antibiotics as well as the Pax5^{+/-} Myd88^{+/-} mice. This would give a lot of clues about the cell intrinsic developmental alterations which accompany these genetic changes. This is definitely outside the scope of this paper but could be useful for the lab to work on in the future.

Point-by-point answer to the reviewer's comments:

Reviewer #1 (Remarks to the Author):

I appreciate that the authors have made significant changes to this manuscript. Importantly, these include the addition of Pax5 model groups in the experiments presented in Figures 1 and 2. These data, obtained over the same time period as the other reported experiments, are essential for the validation of the foundational findings of the manuscript. In my opinion, the manuscript is strengthened by the inclusion of these comparative data and remains an intriguing one that provides evidence of a new immune activity in driving B-ALL progression.

However, the lack of mechanistic insight provided by the current experiments continues to reduce enthusiasm for the manuscript in its current form. While results from additional experiments have been provided, these do not wholly address the need for more detailed explanations for the observed outcomes. Specifically:

1) The lack of data from the use of defined infections or danger signals remains the biggest weakness in this study. In the resubmission cover letter, the authors refer to previous papers showing effects of MyD88 signaling on B-ALL; however, those BCP ALL studies have all involved in vitro exposure to TLR ligands i.e have not incorporated the impact of systemic TLR-mediated immune stimulation on in vivo leukemogenesis. Such models do not shed much light on the balance of mechanisms at play in a Pax5^{+/-} mouse. If, as the authors contend, Myd88 insufficiency mimics the role of infection as a driver of young onset B-ALL, then the exposure of SPF-housed Pax5^{+/-} mice to infectious agents or TLR ligands should generate leukemia. Critically, such experiments would begin to reduce the number of unknown variables in this model, such as the nature of infection, timing of exposure, pathway specificity, age-related immune competence. The need for such insights is demonstrated by the failure of MyD88^{+/-} Pax5^{+/-} double transgenics to show earlier onset of disease than Pax5^{+/-} mice. While an explanation for the late onset is proposed, the fact that all additional deaths occur after 15 months of age in mice with constitutional immune dysregulation raises concerns this is not recapitulating childhood B-ALL leukemogenesis but rather reflecting activities exerted by an old immune system.

This is indeed an important remark. Accordingly, in this new revised version, following the reviewer's advice, we have now included a **new Supplementary Fig. S9** where SPF-housed two months-old WT and Pax5^{+/-} mice were subsequently intraperitoneally (i.p.) injected with PBS or 35 mg LPS (a Myd88-dependent TLR ligand) every other day for eight times over 2 weeks. The exposure of SPF-housed Pax5^{+/-} mice to this Myd88-dependent TLR ligand generated B-ALL in 20% of the mice. On the contrary, the exposure of SPF-housed Pax5^{+/-} mice to Myd88-independent TLR ligand (Poly(I:C)) did not generate B-ALL, therefore arguing against an effect due to aging, and supporting the direct participation of Myd88 modulation in B-ALL development in this model.

Supplementary Fig. S9: *Pax5*^{+/-} mice injected with LPS develop B-ALL without infection exposure. **A)** B-ALL-specific survival of *Pax5*^{+/-} mice injected with LPS (a Myd88-dependent TLR ligand) (orange line, n=10), *Pax5*^{+/-} mice injected with poly(I:C) (a Myd88-independent TLR ligand) (light blue line, n=10), *Pax5*^{+/-} mice injected with PBS (as control)

(green line, n=10), control wild type (WT) mice injected with LPS (red line, n=10), WT mice injected with poly(I:C) (dark blue line, n=10), and WT mice injected with PBS (black line, n=10), all of them housed in an SPF facility (without exposure to common infections). Log-rank (Mantel-Cox) test p-value=0,1464 when comparing *Pax5*^{+/-} mice injected with LPS and *Pax5*^{+/-} mice injected with PBS. **B)** Flow cytometry representative illustration of the percentage of leukemic B cells (B220⁺IgM⁺ subsets) in PB, BM, spleen and LN from a diseased *Pax5*^{+/-} mouse injected with LPS compared to an age-matched healthy WT mouse. **C)** Haematoxylin and eosin staining of a tumour-bearing *Pax5*^{+/-} mouse injected with LPS unexposed to common infections showing infiltrating blast cells in the spleen, liver, and lymph nodes and compared with a healthy WT mouse. Loss of normal architecture can be seen due to the infiltrating cells morphologically resembling lymphoblast. Magnification and the corresponding scale bar are indicated in each case.

2) In contrast to the lack of Myd88-dependent TLR ligands, the authors have added data regarding the acute effects of Poly(I:C) treatment on the “preleukemic” BCP cell population (confusingly, the text on p13 states that the treatment did deplete preleukemic cells, while the figure legend states that it did not. Similar contradictory statements are made for the myeloid enrichment after treatment). This is likely just a typo issue, but it draws attention to the lack of clarity as to what constitutes a preleukemic cell and whether this assay has the resolution to detect the depletion of genuine pre-malignant BCP cells. The increased latency is attributed to “early Myd88-independent TLR ligation”, but in the absence of the paired experiment with a Myd88-dependent ligand, it is impossible to determine how Poly(I:C) extends the disease latency or whether it really induces an outcome distinct from Myd88-dependent TLRs.

In the revised version the typos in the main text related to **Supplementary Figures S26 and S27** (now Figure S27 and S28) have been corrected (page 13).

In the revised version and following the reviewer’s advice, we have now included a **new Supplementary Fig. S9** showing that the exposure of SPF-housed *Pax5*^{+/-} mice to the Myd88-independent TLR ligand Poly(I:C) did not generate B-ALL.

*3) While the inclusion of data from the ETV6-RUNX1 model on Myd88 expression after antibiotic treatment is appreciated, the interpretation of the result is not convincing. It appears from Figure 3A that, although the mean is similar to that of untreated *Pax5* mice, half of the ETV6-RUNX1 mice reduce Myd88 expression to levels seen in *Pax5*^{+/-} Abx mice and *Pax5* leukemia. The inability of Abx-treatment to drive leukemia progression in the ETV6-RUNX1 model, therefore, does not appear to be due to a failure to down-regulate Myd88 expression. While the different leukemia models may have very different reliance on Myd88 signaling, it highlights the need to identify downstream mechanism(s) to provide possible explanations for such differences.*

In the revised version we have now clarified this point (page 9), indicating that “This finding suggested that the inability of Abx-treatment to drive leukemia progression in the ETV6-RUNX1 model therefore, does not appear to be due to a failure to down-regulate Myd88 expression in precursor B-cells. While the Sca1-ETV6-RUNX1 model might have very different reliance on Myd88 signaling or on the cell intrinsic developmental stage where molecular alterations should take place, these results highlight the need to identify downstream mechanisms to provide possible explanations for the inability of Abx-treatment to drive leukemia progression in the ETV6-RUNX1 model.”

Overall, the authors have addressed weaknesses in the initial submission, especially by providing necessary controls. However, in the absence of a deeper investigation of the mechanism underlying Myd88’s influence on leukemia progression (eg whether it is Myd88 signaling within preleukemic BCP or the broader immune system that is the key driver), the study does not provide sufficient insight to merit publication in Nature Communications.

In the revised version we have now included the experiments showing that *in vivo* Myd88 signaling within preleukemic B-cell population is the key driver.

Reviewer #2 (Remarks to the Author):

In this revision, the authors have improved the manuscript throughout. They have addressed all of my previous concerns appropriately.

Reviewer #3 (Remarks to the Author):

The authors have made a lot of effort to improve this manuscript which is quite commendable. Especially the addition of more control groups in the mouse experiments is very heartening since it contributes a lot to scientific rigor. Most of the major queries have now been resolved. However, there are a few more minor comments which would help in polishing the manuscript further.

We have carefully addressed all questions in the revised manuscript, as detailed below.

Figure 1B: The X-axis should ideally begin from 0. This makes the data look spuriously inflated.

Following the reviewer's suggestion, we have modified **Figure 1B** so that x-Axis begins from 0, and we have also assigned the same color to each experimental group so that the entire figure is now more homogeneous.

Figure S3: The legend mentions about the thymus being infiltrated which is not seen in the flow cytometry data; The data about C-kit and CD25 positivity mentioned in the legend is not seen in the graphs.

We acknowledge the reviewer for this comment, the figure's legend was indeed confusing and we have now modified it to clarify the figure as follows:

“Supplementary Fig. S3: Flow cytometric analysis of hematopoietic subsets in diseased Pax5^{+/-} early-exposure mice. Representative plots of cell subsets from the

thymus, spleen, bone marrow (BM), peripheral blood (PB), and lymph nodes (LN) are shown from a diseased Pax5^{+/-} early-exposure mouse. A total of 7 diseased mice were analyzed by flow cytometry (age: 9-17 months). FACS analysis revealed a cell surface phenotype CD19^{+/-} B220⁺IgM⁻ for tumor cells that extended through BM, PB, spleen, and LN.”

Figure S6: The early exposure groups (including the WT) show a significant decrease in PB B-cells. The authors should speculate a bit on this phenomenon; esp. the decrease in the WT mice.

In page 7 of the revised version we now indicate: “The early exposure to common pathogens induced a significant decrease in PB B-cells in both WT and Pax5^{+/-} mice. However, this decrease does not seem to be related to leukemogenesis as WT mice never develop B-ALL”.

Figure S8: This data is still too haphazard to me with large error bars in most of the samples and genes. The authors can try to plot the data using a scatter plot or plot only the relative expression instead of fold changes to make it clearer.

We appreciate the criticism, and we have tried to represent the data in the formats suggested by the reviewer. However, due to the relatively limited number of samples available and the noise inherent to the technology, other formats of data representation show the same problems with the error bars, and are less visually understandable. Therefore, we have considered clearer to maintain the current figure as it is.

Figure 3 legend: ‘Development’ instead of ‘emergency’

In the revised version we have replaced “emergency” by “development” in Figure 3 legend.

Figures S14-S15: Did the authors find any upregulated signature for B-ALL subtype when they compared the transcriptome of leukemic Pax5^{+/-} Myd88^{WT} with the Pax5^{+/-} Myd88^{+/-} cells?

Yes, we found upregulated signatures for the human BCR-ABL⁺ B-ALL subtype when we compared the transcriptome of leukemic Pax5^{+/-} Myd88^{WT} with the Pax5^{+/-} Myd88^{+/-} cells. Now it is included as a new panel within Figure S16 (new Figure S16B).

B

Since the authors compared the human transcriptomic signatures with mouse transcriptome, the methodology can be elaborated for the use of readers, maybe in the supplementary information.

Now, the list of gene sets used to compare the human transcriptomic signatures with mouse transcriptome is included in **Supplementary Table S11**.

Also, the figures showing the NES and FDR can be represented as bubble plots which can also show enriched gene count along with ES and FDR/p-value (S14-15. S20-21).

Following the reviewer's advice, we have now remade Figures (S14-15. S20-21: **now Figures S15-16. S21-22**) as bubble plots showing enriched gene count along with ES and FDR/p-value and exemplify in the next figure:

Figure S19: Can add a composite Venn diagram with DE gene numbers to get an idea of the degree of overlap between groups.

We thank the reviewer for this suggestion. We have now added (**new Figure S20D**) a Venn diagram showing the overlapped genes of the differentially expressed genes between the three groups analyzed.

Figure S22, S24 and S25: Are not legible. Can be redrawn clearly and text to be added in English.

We apologize for this mistake. Figures S22, S24 and S25 have now been corrected in the revised version of the manuscript (now Figures S23, S25 and S26).

Figure S26: The authors can try to plot absolute numbers of cells instead of the percentages if they have the bone marrow cell counts. This may provide better translational value than the percentages alone.

In the revised version and following reviewer's advice, we have plotted absolute numbers of cells in Figure S26 (now Figure S27) instead of the percentages.

On a related note, there definitely appears to be some B-cell developmental changes in the

bone marrow. The authors can try to study Hardy Fractions in the bone marrow of the Pax5^{+/-} exposed to infection/antibiotics as well as the Pax5^{+/-} Myd88^{+/-} mice. This would give a lot of clues about the cell intrinsic developmental alterations which accompany these genetic changes. This is definitely outside the scope of this paper but could be useful for the lab to work on in the future.

We would like to thank the reviewer for this suggestion that we plan to follow up.

REVIEWERS' COMMENTS

Reviewer #1 (Remarks to the Author):

The authors have addressed, to varying degrees, each of my previous comments. Personally, I would move the LPS figure to the body of the manuscript, but perhaps I am biased. While it still leaves many questions unanswered, this version presents a better balance of its novel findings and limitations. The results will be of considerable interest to the field.

Reviewer #3 (Remarks to the Author):

In this revised version, the authors have improved the manuscript further. They have addressed all the concerns appropriately.

Point-by-point answer to the reviewer's comments:

Reviewer #1 (Remarks to the Author):

The authors have addressed, to varying degrees, each of my previous comments. Personally, I would move the LPS figure to the body of the manuscript, but perhaps I am biased. While it still leaves many questions unanswered, this version presents a better balance of its novel findings and limitations. The results will be of considerable interest to the field.

Following the reviewer's suggestion, we have moved the LPS figure to the body of the manuscript (**new Figure 4**).

Reviewer #3 (Remarks to the Author):

In this revised version, the authors have improved the manuscript further. They have addressed all the concerns appropriately.